# ANKRD1 is a mesenchymal-specific driver of cancer-associated fibroblast activation bridging androgen receptor loss to AP-1 activation

Luigi Mazzeo[1,2], Soumitra Ghosh[3], Emery Di Cicco[2], Jovan Isma[1], Daniele Tavernari [4,5,6], Anastasia Samarkina [1], Paola Ostano [7], Markus K. Youssef [1], Christian Simon[3,8] & G. Paolo Dotto [1,2,3,8] ✉

There are significant commonalities among several pathologies involving fibroblasts, ranging from auto-immune diseases to fibrosis and cancer. Early steps in cancer development and progression are closely linked to fibroblast senescence and transformation into tumor-promoting cancer-associated fibroblasts (CAFs), suppressed by the androgen receptor (AR). Here, we identify ANKRD1 as a mesenchymal-specific transcriptional coregulator under direct AR negative control in human dermal fibroblasts (HDFs) and a key driver of CAF conversion, independent of cellular senescence. ANKRD1 expression in CAFs is associated with poor survival in HNSCC, lung, and cervical SCC patients, and controls a specific gene expression program of myofibroblast CAFs (my-CAFs). ANKRD1 binds to the regulatory region of my-CAF effector genes in concert with AP-1 transcription factors, and promotes c-JUN and FOS association. Targeting ANKRD1 disrupts AP-1 complex formation, reverses CAF activation, and blocks the pro-tumorigenic properties of CAFs in an orthotopic skin cancer model. ANKRD1 thus represents a target for fibroblast-directed therapy in cancer and potentially beyond.

Identification of common molecular mechanisms underlying fibroblast changes in a spectrum of diseases, such as autoimmune-diseases, fibrosis, and cancer, could be of great clinical importance[1,2]. While cellular senescence suppresses the malignant potential of cancer cells through irreversible withdrawal from the cell cycle, the same process in surrounding stromal fibroblasts can promote early steps of carcinogenesis through the induction of several genes with pro-inflammatory and matrix remodelling functions, constituting the so-called *Senescence Associated Secreted Phenotype* (SASP)[3].

Expression of SASP genes is also a key feature of fully established cancer-associated fibroblasts (CAFs)[4–8], which have, however, escaped molecular mechanisms of cellular senescence[9] and can co-evolve and expand with cancer cells[10]. While the transcriptional program leading to senescence and SASP induction has been under intensive investigation[3], the identification of mechanism(s) that are selectively involved in CAF activation without impinging on senescence would be of substantial interest to counteract the whole cancer process.

[1]Department of Immunobiology, University of Lausanne, Epalinges, Switzerland. [2]Cutaneous Biology Research Center, Department of Dermatology, Massachusetts General Hospital and Harvard Medical School, Charlestown, MA, USA. [3]ORL service, Centre Hospitalier Universitaire Vaudois, Lausanne, Switzerland. [4]Department of Computational Biology, University of Lausanne, Lausanne, Switzerland. [5]Swiss Cancer Center Léman, Lausanne, Switzerland. [6]Swiss Institute of Bioinformatics, Lausanne, Switzerland. [7]Cancer Genomics Laboratory, Edo and Elvo Tempia Valenta Foundation, Biella 13900, Italy. [8]International Cancer Prevention Institute, Epalinges, Switzerland. ✉e-mail: gdotto@mgh.harvard.edu

UVA exposure is a significant cause of skin aging[11] and cancer[12]. It can directly target dermal cells and suppress the expression of critical negative regulators of dermal fibroblast senescence and CAF activation, such as the androgen receptor (AR)[13] and CSL (RBP-Jκ), the effector of canonical Notch signaling endowed with intrinsic transcription repressive function[9,14]. AR and CSL physically associate and converge on direct suppression of many SASP and CAF effector genes as well as the senescence effector *CDKN1A* while, at the same time, interacting and suppressing p53 activity[9,13]. As a result, loss or down-regulation of either AR or CSL gene leads to p53-dependent cellular senescence and SASP induction, with compromised p53 activity leading to full CAF activation[9,13]. In contrast to these negative regulators, a multiplicity of transcription factors has been implicated as positive determinants of CAF activation, as endpoints of converging signalling pathways[4–7,15,16].

CAFs consist of heterogeneous populations with distinct pro-inflammatory or pro-fibrotic properties that can be differentially distributed even within individual tumors[1,2]. Taking a "reverse" functional approach, we have previously shown that heterogeneity of CAF populations can be recapitulated by differential modulation of the fibroblast growth factor (FGF) versus transforming growth factor-beta (TGFβ) signaling pathways, with the ETV1 and SMAD transcription factors, respectively, being involved[17,18]. In concert with these and other transcription factors, members of the activated protein-1 (AP-1) family are likely to play an important role in CAF activation, which goes along their highly studied role in a variety of other cellular contexts[15,19–21].

A key question is whether mesenchymal-specific transcriptional determinants of CAF activation can be identified that interplay with the above regulators and can be targeted to suppress the process. By a combination of experimental and bioinformatic approaches, we have found that ANKRD1, a transcriptional coactivator so far mostly studied for its role in cardiac development and pathology[22,23], and implicated by a mouse genetic study in skin wound healing[24] is a key determinant of CAF activation. We find that the *ANKRD1* gene is specifically expressed and upregulated in CAFs of various cancer types, is a direct target of AR suppression, and is both required and sufficient for CAF activation. Moreover, ANKRD1 expression was considerably elevated in fibroblast-related pathologies such as hypertrophic scarring, keloids, and idiopathic pulmonary fibrosis, pointing to a likely common link with CAF activation. The ANKRD1 protein physically associates with two key AP-1 family members, c-JUN and FOSL2, promoting their heterodimer formation and CAF gene upregulation. Targeting ANKRD1 by FANA-modified antisense oligonucleotides (ASOs)[25] blocks the AP-1 axis and reverts the CAF phenotype hampering the pro-tumorigenic properties of these cells.

## Results

### ANKRD1 is a mesenchymal-specific CAF marker
Identifying mesenchymal-specific regulatory mechanisms targeting CAF activation could be of translational importance for stroma-directed cancer therapy, with potential implications for the treatment of other fibroblast-related diseases. Towards this goal, we started by comparing the global transcriptomic profiles of (i) multiple CAF strains derived from skin squamous cell carcinomas (SCCs) versus matched normal fibroblasts (NFs) of surrounding unaffected skin;[17] (ii) the profiles of human dermal fibroblasts (HDFs) plus/minus androgen receptor (*AR*) gene silencing, which elicits early steps of CAF activation;[13] (iii) the profiles of different CAF strains plus/minus treatment with the bromodomain and extra terminal protein (BET) inhibitor JQ1, which can restore AR expression and reverts the CAF phenotype[26]. This comparative analysis pointed to *ANKRD1* as the top gene with transcription regulatory functions that was consistently (i) upregulated in CAFs versus matched NFs, (ii) induced in HDFs by *AR* gene silencing, and (iii) repressed in CAFs by treatment with JQ1 (Fig. 1a, Supplementary Data file 1).

RT-qPCR, immunoblot, and immunofluorescence (IF) analysis showed that the levels of ANKRD1 expression are markedly increased in a set of early passage CAFs derived from skin SCCs versus human dermal fibroblasts (HDFs) from flanking skin of the same patients (Fig. 1b–d). Concordantly elevated expression of CAF marker genes was found in CAF versus HDF strains by global transcriptomic analysis of the same RNA samples assessed for *ANKRD1* levels (Supplementary Fig. 1a) and/or was previously reported[10,18]. Stable differences in expression levels of *ANKRD1* and *ACTA2* were confirmed by RT-qPCR analysis of some CAF versus HDF strains at 1–2 later passages (Supplementary Fig. 1b).

Elevated ANKRD1 expression was also found in melanoma-derived CAFs versus a reference panel of HDFs (Fig. 1c, lower panel). RT-qPCR analysis also showed drastically higher expression of *ANKRD1* in CAFs than in multiple SCC cell lines, in most of which it was below the detection limit (Supplementary Fig. 1c), consistent with the selective ANKRD1 expression in cells of the mesenchymal lineage, mostly cardiomyocytes[22,23]. Elevated *ANKRD1* expression levels were found by immunofluorescence-guided laser capture microdissection (LCM) and RT-qPCR analysis of fibroblasts (PDGFR-α positive cells) associated with multiple skin SCCs (CAFs) versus fibroblasts of flanking skin (NF) (Fig. 1e).

Immunofluorescence (IF) analysis of skin SCC samples further demonstrated elevated ANKRD1 expression in CAFs versus fibroblasts of flanking skin with no expression in cancer cells (Fig. 1f, Supplementary Fig. 1d, e). These findings are not limited to skin CAFs as analysis of gene expression profiles of CAFs from breast[27], lung[28], and colorectal cancer[29] showed significantly higher *ANKRD1* expression relatively to matched fibroblasts from the same patients (Fig. 1g). Up-regulated *ANKRD1* expression was also found in expression profiles of other fibroblast-related pathologies such as hypertrophic scarring[30], keloids[30], and idiopathic pulmonary fibrosis[31] (Fig. 1h).

Thus, *ANKRD1* is a mesenchymal-specific gene with a transcription regulatory function markedly up-regulated in CAFs.

### ANKRD1 is under direct negative control of AR
AR functions as a negative regulator of the early steps of CAF activation[13]. Chromatin immunoprecipitation (ChIP) analysis with anti-AR antibodies combined with Tn5 transposase tagging ("ChIPmentation";[32] and sequencing showed two major AR binding peaks to the promoter region of the *ANKRD1* gene (5 Kb upstream of the transcription start site) (Fig. 2a). Direct ChIP assays with an AR antibody in multiple HDF strains confirmed binding of AR to the sites identified by ChIPmentation-seq (Fig. 2b).

Unbiased analysis of public ChIP-seq experiments (CISTROME, http://dbtoolkit.cistrome.org/)[33] identified AR among the most represented transcription factors binding to the *ANKRD1* gene (Fig. 2c). A similar analysis focusing on one of the AR binding peaks (site1) identified by ChIPmentation-seq in HDFs showed highly significant binding of AR also in the CISTROME database (Fig. 2d). These findings are of functional significance, as we found consistent and strong induction of *ANKRD1* expression in several HDF strains plus/minus AR silencing by both RT-qPCR and western blot analysis (Fig. 2e, f). Induction of *ANKRD1* expression, in parallel with the *COL1A1* and *HAS2* CAF marker genes, was also observed after treatment of HDFs with UT-155[34], a small-molecule AR inhibitor causing the degradation of AR protein (Fig. 2 g, h). Conversely, treatment of multiple CAF strains with the BET inhibitor JQ1, which reverses CAF activation at the same time as it restores AR levels[13], suppressed *ANKRD1* expression (Fig. 2i).

Thus, *ANKRD1* is a target of AR transcriptional repression induced by loss of AR function as occurring in HDF to CAF conversion.

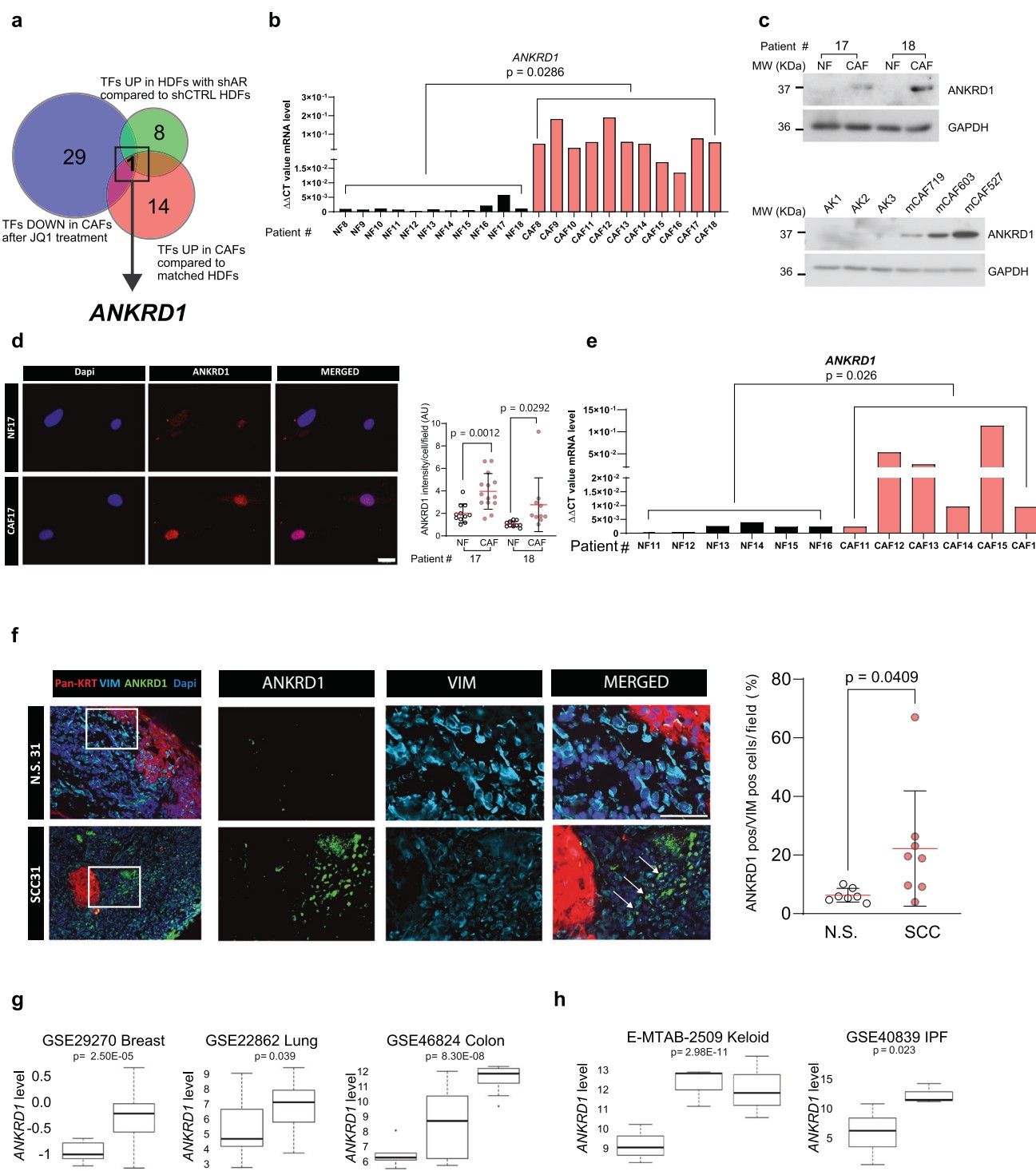

## Sustained *ANKRD1* expression is required for a CAF transcription program of clinical significance

We employed a loss-of-function approach to assessing further the functional significance of elevated *ANKRD1* expression in CAFs. Silencing of the *ANKRD1* gene by two different shRNA lentiviral vectors in multiple CAF strains reduced the expression of several CAF marker genes, such as *ACTA2, HAS2, COL1A1* (Fig. 3a, Supplementary Fig. 2a), while not affecting the overall proliferation of cells, as assessed by measuring the Ki67 proliferative index (Fig. 3b, Supplementary Fig. 2b). In coculture assays on thin Matrigel layer[13], the expansion of SCC cells (FaDu) was reduced in the presence of CAFs with silenced *ANKRD1* versus control (Fig. 3c). Silencing of *ANKRD1* resulted in

consistent morphological changes of CAFs, with an elongated shape and reduced surface occupation (Supplementary Fig. 3a). As with CAFs in isolation, quantification by ImageJ analysis showed that *ANKRD1* silencing causes no significant changes in fibroblast density as opposed to cancer cells (Supplementary Fig. 3b).

Sphere formation of cutaneous SCC13 and FaDu cells in a 3D assay in Matrigel and their ability to invade in an organoid model were similarly reduced in the presence of CAFs with silenced *ANKRD1* (Fig. 3d, e). Double IF analysis with anti-KERATIN and anti-VIMENTIN antibodies for cell type identification showed close intermingling of SCC cells and CAFs, both under control conditions and with silenced *ANKRD1* (Fig. 3e; Supplementary Fig. 3c). In an orthotopic model of

**Fig. 1 | ANKRD1 is a mesenchymal-specific cancer-associated fibroblast (CAF) marker. a** Differentially expressed transcription factors and co-factors (TFs) in published data sets. (1) TFs upregulated (UP) in cancer-associated fibroblasts (CAFs, red, GSE122372), and present in Animal Transcription Factor database (http://bioinfo.life.hust.edu.cn/AnimalTFDB/). (2) TFs downregulated (DOWN) in CAFs treated with BET inhibitor JQ1 (blue, GSE81406). (3) TFs upregulated (UP) in human dermal fibroblasts (HDFs) following *AR* silencing (green, GSE107321). For all data-sets: logFC > 2, *p* < 0.05 two-way ANOVA test. **b** RT-qPCR analysis of *ANKRD1* expression in skin-SCC patient-derived CAFs and matched normal fibroblasts (NFs) (#8 to #18), amplification cycle normalized to *RPLPO* (ΔΔCT). *n*(strains) = 11, two-tailed unpaired t-test. **c** ANKRD1 and GAPDH Immunoblot analysis (WB). Top: one experiment including 2 patient's CAFs and matched HDFs *n*(strains) = 4. Bottom: second independent experiment with melanoma-derived CAFs and unmatched HDFs *n*(strains) = 6. **d** Representative immunofluorescence (IF) images of ANKRD1 (red), DAPI (blue) in CAFs and matched NFs. *n*(strains) = 2, *n*(fields) = 11 for NF17 and NF18, 14 for CAF17, 10 for CAF18, mean ± SD, two-tailed unpaired t-test. Scale bar: 20 μm. **e** RT-qPCR analysis of *ANKRD1* mRNA in Laser capture microdissection

(LCM) samples from patients' stroma underlying and flanking in situ skin SCC lesions. Amplification cycles were normalized to *ACTB* (ΔΔCT). *n*(strains) = 6, two-tailed unpaired t-test. **f** ANKRD1 (green), Pan-KRT (red) and VIMENTIN (blue) IF in skin SCC and matched normal skin (NS). *n*(fields) = 7 for normal, and 8 for SCC, mean ± SD, two-tailed paired t-test with Mann-Whitney correction. Scale bar: 50 μm. The experiment was repeated in other 2 SCC tissues as reported in Supplementary Fig. 1e. **g, h** *ANKRD1* expression in CAFs versus NFs from breast (GSE29270, *n* = 63), lung (GSE22862, *n* = 30), colon (GSE46824, *n* = 34), normal fibroblasts (NF), *n* = 9, primary tumor CAF (P-CAF), *n* = 14, metastatic CAF (M-CAF), *n* = 11. *ANKRD1* expression in fibroblasts from hypertrophic scar and keloids (E-MTAB-2509, *n* = 27), normal fibroblasts = 9, hypertrophic fibroblasts = 9, keloid fibroblasts = 9, and from idiopathic pulmonary fibrosis (IPF) (GSE40839, *n* = 21), control fibroblasts = 10 and IPF = 11. Moderated t-statistic for two-class comparison. Moderated F-statistic for three-group comparison with Benjamini-Hochberg correction. Minimum value, first quartile (lower box bound), median (centre), third quartile (top box bound), maximum value. Whisker length is a maximum of 1.5 times the inter-quartile range. Points represent values outside intervals.

SCC based on the intradermal injection of admixed SCC and fibro-blasts, the lesions formed by SCC cells together with CAFs with silenced *ANKRD1* were significantly smaller than those with control CAFs, with lesser tumor cell density (Fig. 3f, g). Double IF (immuno-fluorescence) approach with anti-vimentin and anti-keratin antibodies was used to identify cell types. Additionally, we used a human-specific anti-lamin A/C antibody to distinguish human from mouse cells. Consistent with previous findings[35–37], we observed that a significant portion of cancer-associated fibroblasts (CAFs) at the end of the experiment were replaced by mouse cells (Supplementary Fig. 3d). Additional in vivo work needs to be addressed to understand the dynamics of the CAF's contribution on tumor growth, specifically addressing early vs late time points.

Underlying these findings, global transcriptomic analysis showed a battery of genes that were consistently down- or up- regulated in four different CAF strains with *ANKRD1* silencing versus controls (Fig. 4a, Supplementary Data file 2). Gene Set Enrichment Analysis (GSEA) showed that a CAF gene signature extracted from skin CAFs versus matched HDFs[17] was strongly enriched in the transcriptomic profiles of CAFs with control versus silenced *ANKRD1*, with a similar association with a gene signature of myofibroblastic CAFs (myCAFs) and an inverse relation with one of inflammatory CAFs (iCAFs)[38] (Fig. 4b). Enrichment for several other gene signatures related to fibroblast activation was also strongly suppressed by *ANKRD1* silencing (Fig. 4c).

Analysis of the genes consistently downmodulated by *ANKRD1* silencing in all four tested CAF strains (FC < −2 fold, *p* value < 0.05) by the ENRICHR/ARCHS4 webtool[39] showed that more than 50% (269 out of 512) are specific for mesenchymal cell lineages (fibroblasts, myo-fibroblasts, or smooth muscle cells; Fig. 4d). A consensus gene signature based on these genes (Supplementary Data file 3), which has a minor overlap (<3%; or 9 genes) with an established EMT signature (GO:0001837) (Supplementary Fig. 3e), was used to probe into pub-lished single-cell RNA-seq profiles of Head/Neck SCCs[40]. The signature was found to be specific for the fibroblasts versus other mesenchymal cell populations (B cell, T cell, myocyte, endothelial cell, cancer cell, macrophage, mast cell, and dendritic cell) present in the tumors (Fig. 4e) and, within cells of the fibroblast lineage, to positively cor-relate with those expressing our CAF signature (Fig. 4f).

To probe into the clinical significance of the findings, we exam-ined the expression profiles of several cancer cohorts. A positive cor-relation was found in lung (LUSC), cervical (CESC) and head/neck (HNSC) SCCs of the TCGA database (https://portal.gdc.cancer.gov/) between levels of *ANKRD1* expression and the mesenchymal *ANKRD1*-dependent gene signature that we established (Fig. 4g), which, by the EPIC (Estimating the Proportions of Immune and Cancer cells[41]) tool, was strongly associated with the estimated proportion of CAFs in tumors (Supplementary Fig. 3f). Importantly, Kaplan Meier's analysis

showed that levels of *ANKRD1* expression across examined SCC types are associated with patients' poor survival (Fig. 4h).

Thus, sustained *ANKRD1* expression is required to maintain a CAF phenotype and a transcriptional program of clinical significance.

### Increased ANKRD1 expression triggers CAF activation

For further mechanistic insights, we assessed the consequences of increased ANKRD1 expression in primary HDFs, in the absence of the changes that occur in CAFs. Infection of multiple HDF strains with an *ANKRD1*-expressing lentivirus resulted in increased ANKRD1 protein levels comparable to those found in CAFs or HDFs with *AR* gene silencing (Fig. 5a). Unlike the effects induced by *AR* gene silencing[13], increased ANKRD1 expression caused no morphological changes associated with cellular senescence and no decrease in cell prolifera-tion (as assessed by EdU incorporation assay (Supplementary Fig. 4a, b), and no induction of the senescence effector gene *CDKN1A* (Fig. 5b). By contrast, CAF marker genes such as *ACTA2*, *COL1A1*, and *INHBA*, were consistently induced by increased *ANKRD1* expression in multiple HDF strains (Fig. 5b). Importantly, ANKRD1-overexpressing HDFs enhanced the expansion of both cutaneous and oral SCC cells in 2D coculture assays (Supplementary Fig. 4c, d) and, in a Matrigel-based 3D assay, sphere-forming capability (Supplementary Fig. 4e). In vivo intradermal injection assays showed that SCC cells formed larger tumors with greater cellularity when admixed with HDFs with ANKRD1 overexpression versus controls (Supplementary Fig. 4f, g).

At the transcriptomic level, a large set of genes was found to be consistently modulated by *ANKRD1* overexpression in multiple HDF strains (Fig. 5c; Supplementary Data file 4). Mirroring the con-sequences of silenced *ANKRD1* in CAFs, Gene Set Enrichment Analysis (GSEA) showed a positive association of the profiles of ANKRD1 over-expressing HDFs with several CAF gene signatures from skin and Head/Neck SCC, breast, colon, and lung cancer[17,28,42–44] (Fig. 5d, Supple-mentary Fig. 5a). In addition, a myCAF and a TGFβ gene signatures were positively enriched in the profiles of ANKRD1-overexpressing HDFs, while there was an inverse relation with an iCAF signature[38,45] (Fig. 5e, Supplementary Fig. 5b).

Loss or downmodulation of AR expression in HDFs triggers a global program of CAF activation with multiple makers of both my-CAF and i-CAF populations (Supplementary Fig. 5c)[13]. A signature of up-regulated genes in HDFs with *AR* gene silencing was used for GSEA of profiles of HDFs plus/minus ANKRD1 overexpression. As shown in Fig. 5f, we found a bimodal distribution of the AR-dependent gene signature, with approximately half being positively associated with the profiles of ANKRD1 overexpressing HDFs and half being inversely related. GO analysis showed that the group of AR-dependent genes positively associated with ANKRD1 overexpression are enriched for genes involved in TGF-ß signaling, while the second is related to pro-

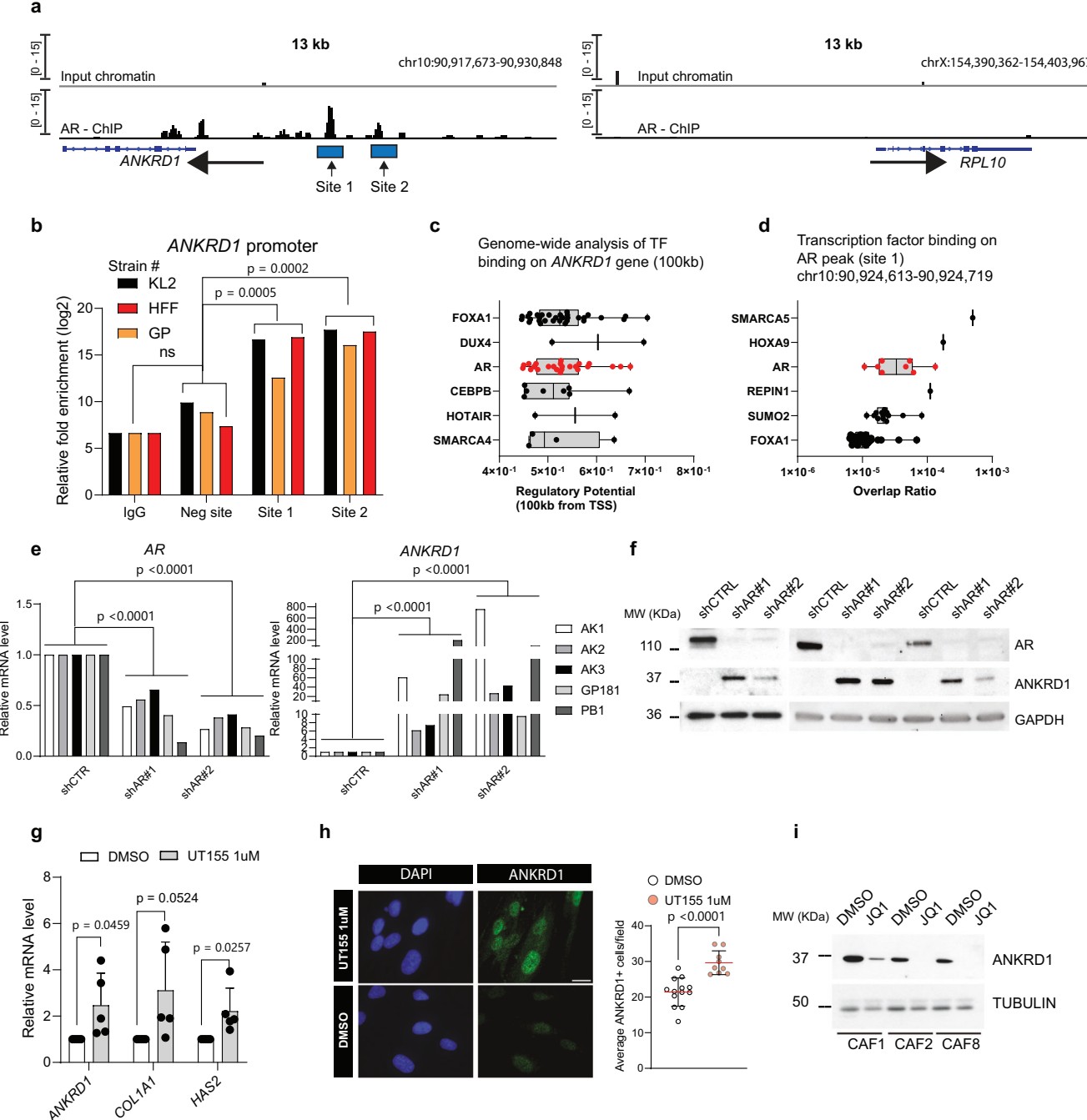

**Fig. 2 | ANKRD1 is under direct negative control of AR. a** Left: AR-binding peaks (BEDGRAPH) to the *ANKRD1* gene in human primary foreskin fibroblasts (HFF = SG1) by ChIPmentation sequencing. Shown is the IGV representation of 13 KB region encompassing the *ANKRD1* gene. Right: *RPL10* gene was used as negative control for AR binding, no binding peaks were detected. **b** Validation of AR binding peaks shown in **a** in three HDF strains to the *ANKRD1* promoter versus a negative site (exon 8 of *ANKRD1*) by ChIPmentation RT-qPCR. AR binding was quantified relative to non-immune IgGs via qPCR amplification. *n*(strains) = 3, two-way ANOVA. **c** Global prediction of transcription factor binding using the Cistrome DB toolkit (http://dbtoolkit.cistrome.org/). Regulatory potential (RP) scores for indicated transcription factors are represented with box plots, with boxes showing the interquartile range, center line representing the median value; minimum and maximum values delineate the range of data points. *n*(total ChIP datasets analyzed) = 200, top 6 TF are shown. Each point represents a separate ChIP-seq dataset. **d** ChIP-seq data of transcription factors showing the highest overlap ratio in the genomic region bound by AR (Site 1) using the CistromeDB toolkit. AR and other transcription factors are represented using box plots, with boxes showing the

interquartile range, center line representing the median value; minimum and maximum values delineate the range of data points. *n*(total ChIP datasets analyzed) = 83, top 6 TF are shown. Each point represents a separate ChIP-seq dataset. **e** RT-qPCR analysis of different HFF strains with or without *AR* silencing using two different shRNAs (shAR#1, shAR#2) versus shRNA control (shCTRL). Fold change (FC) relative to shCTRL, mRNA normalized to *RPLPO*. *n*(strains) = 5. Two-way Anova with Dunnett's multiple comparison's correction. **f** WB for ANKRD1 and AR in HDFs with *AR* silencing (shAR#1 and shAR#2) compared to control HDFs (shCTRL). Anti-GAPDH was used as a loading control. *n*(strains) = 3. **g** RT-qPCR analysis of UT-155-treated HDFs (1 μM, 48 h) compared to DMSO-treated HDFs; expressed as relative FC compared to DMSO. mRNA levels are normalized to *RPLPO*. *n*(biological replicates) = 5, mean ± SD, unpaired t-test. **h** IF of ANKRD1 (green), DAPI (blue) in UT-155-treated HDFs (1 μM, 48 h) compared to DMSO-treated HDFs. *n*(fields) = 13 (DMSO), 9 (UT-155), mean ± SD, unpaired two-tailed t-test. Scale bar: 20 μm. Data points indicate the average number of cells/fields from three independent experiments. **i** WB of ANKRD1 and TUBULIN in patient-derived JQ1-treated CAFs (0.5 μM, 48 h) versus DMSO treatment. *n*(strains)=3.

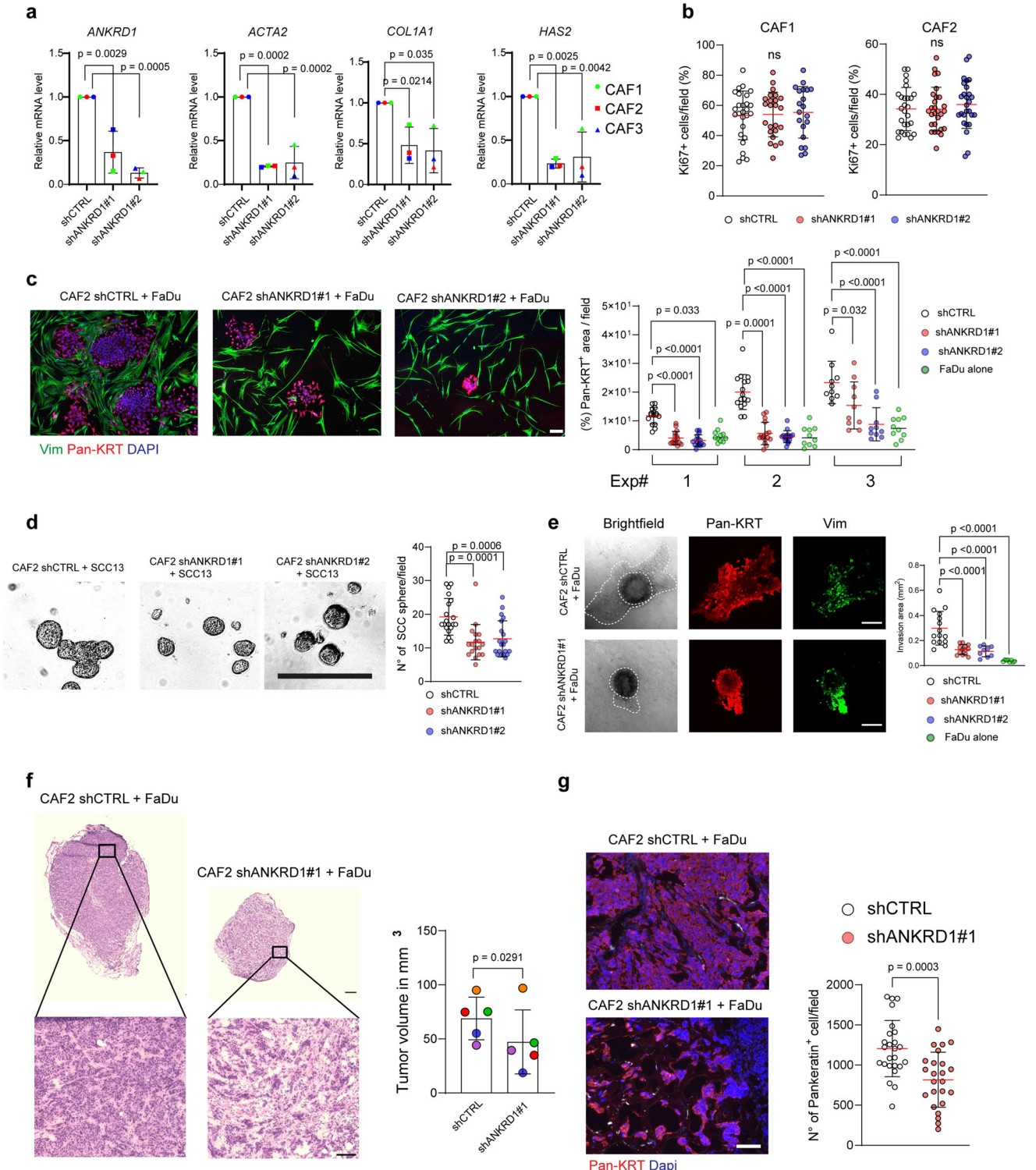

inflammatory cytokines and interferon (IFNs) signaling (Supplementary Data file 5). The results were further validated by analysis of single-cell RNA-seq profiles of H/N SCCs[40] with the ANKRD1 signature score (top 250 genes upregulated by ANKRD1 overexpression FC > 2 p value < 0.05, Supplementary Data file 4) being enriched in the myoCAF subpopulation (Fig. 5g).

Emerging evidence points to a likely common basis of CAF activation and other fibroblast-related disorders[1,2]. GSEA with a signature of upregulated genes in pulmonary fibrosis (D011658)[46] showed a strong enrichment in the profiles of ANKRD1 overexpressing fibroblasts (Fig. 5h). Single cell RNA-seq analysis of activated fibroblasts

derived from another set of patients affected by pulmonary fibrosis[47], showed an increased ANKRD1 signature score compared to fibroblasts derived from healthy lung fibroblasts (Fig. 5i). Specifically, the ANKRD1 signature score was significantly enriched in a diseased-unique population of fibroblasts (HAS1[+]) characterized by an ECM-remodeling phenotype (Fig. 5i).

As a transcription co-factor, ANKRD1 is not predicted to bind DNA directly. To identify specific target chromatin regions, we employed a modified ChIP-seq technique based on an additional protein-protein cross-linking step to pull down DNA-associated transcriptional complexes[48,49]. More than 40000 binding peaks were obtained and

**Fig. 3 | ANKRD1 is required for cancer-associated fibroblast (CAF) maintenance. a** RT-qPCR of the indicated genes of patient-derived CAFs infected with *ANKRD1*-targeting (shANKRD1#1 and #2) and control shRNA (shCTRL), relative to shCTRL and expressed as amplification cycle thresholds normalized to *RPLPO*. n(strains) = 3, mean ± SD, one-way ANOVA with Dunnett's multiple comparisons test. **b** Immunofluorescence analysis of Ki67 in two strains of primary CAFs after *ANKRD1* silencing, shown as the percentage of Ki67 positive cells per field. Number of fields for shCTRL in CAF1 (=27) and CAF2 (=27), for shANKRD1#1 in CAF1 (=24), and CAF2 (=28), for shANKRD1#2 in CAF1 (=20) and CAF2 (=27). Mean ± SD. One-way ANOVA with Dunnett's multiple comparisons test. Ns=non-significant. **c** Growth enhancing activity of CAFs infected with *ANKRD1*-targeting (shANKRD1#1 and #2) and control shRNA (shCTRL) on neighboring SCC cells (FaDu) by co-culture assays. Immunofluorescence for Pan-keratin (Pan-KRT; for FaDu cells) and Vimentin (VIM; for CAFs). n(independent experiments) = 3, n(fields) for shCTRL, shANKRD1#1 and shANKRD1#2: 16 (Exp#1 and Exp#2), 10 (Exp#3). n(fields) for FaDu: 11 (Exp#1), 10 (Exp#2 and Exp#3), mean ± SD. One-way ANOVA with Holm-Šídák's multiple comparisons test. Scale bar: 100 μM. **d** Representative phase contrast images of spheroid formation. CAFs infected with *ANKRD1*- targeting and control shRNA (shCTRL) were co-cultured with SCC13 cells on Matrigel-coated plates. Double IF analysis with anti-keratin and -vimentin antibodies. Mean ± SD, n(fields) = 20 (shCTRL), 18 (shANKRD1#1), 20 (shANKRD1#2), One-Way ANOVA with Dunnett's multiple comparison's test. Scale bar: 500 μm. Data point show the total fields imaged, from 4 independent experiments. **e** Organoid invasion assay of admixed CAFs infected with ANKRD1- targeting and control shRNA (shCTRL) and SCCs. Representative bright field images, and VIMENTIN and Pan-KRT IF analysis. Quantification of the invasion area was measured as the difference between the core and the invading area, delimited with dotted lines. n(fields) = 15 (shCTRL), 13 (shANKRD1#1), 12 (shANKRD1#2), 8 (Fadu), mean ± SD, One-way ANOVA with Dunnett's multiple comparisons test. Scale bar: 200 μm. Data point show the total fields imaged, from 2 independent experiments. **f** Representative images of H&E staining of back lesions formed by FaDu cells co-injected with CAF#2 cells infected with either shANKRD1#1 or shCTRL vectors in contralateral mouse back skin. n(mice) = 5, mean ± SD. Orange dot: outlier identified by Grubbs test α < 0.1, two-tailed unpaired t-test, p = 0.0291. Scale bar: 500 μm, higher magnification: 100 μm. **g** Altered FaDu cell density and proliferation were quantified as the number of Pan-KRT positive cells per field. n(mice) = 5. Data points show the n(fields analyzed) = 25 for Control and n(fields analyzed) = 24 for shANKRD1. Mean ± SD, unpaired two-tailed t-test. p = 0.0003. Scale bar: 100 μm.

annotated, mostly at promoter regions, followed by distal intergenic regions (Fig. 5j, Supplementary Fig. 5d). By combining transcriptomic and ChIPseq profiles, we identified a large number of upregulated genes (1086) that are direct ANKRD1 targets (Fig. 5k), which, by GSEA, were found to be strongly enriched in profiles of clinically derived CAFs versus matched HDFs (Fig. 5k).

For more detailed insights, by IGV software alignment[50], we focused on loci of several CAF marker genes induced by ANKRD1 overexpression. As shown for the *ACTA2*, *HAS2*, and *COL1A1* genes, ANKRD1-binding peaks coincided with promoter and enhancer regions as identified by H3K27ac binding in HDFs of the Encode database (Fig. 5l).

Thus, increased ANKRD1 expression in HDFs is sufficient to induce CAF activation through direct targeting of CAF effector genes.

## ANKRD1 regulates CAF activation through AP-1 interaction

To identify transcription factors functioning in concert with ANKRD1, we performed motif analysis of the global profile of ANKRD1 binding peaks using the Multiple Em for Motif Elicitation (MEME, https://meme-suite.org/meme/index.html). The most enriched binding sequences in ANKRD1 ChIP-seq peaks were for the Activator Protein 1 (AP-1) complex, followed by a lesser association with those for TEAD and ETV transcription factors, which have a well-established connection with AP-1[21,51,52] (Fig. 6a, Supplementary Data file 6). The findings were complemented by a comparative analysis of the ANKRD1 binding profiles to CAF effector genes versus published ChIP-seq profiles available in the Cistrome DB database (http://dbtoolkit.cistrome.org/). c-JUN, JUND, and several FOS family members were among the top-ranked transcription factors, with a high score of overlapping binding peaks with ANKRD1 in multiple studies (Fig. 6b).

An attractive possibility was that ANKRD1 could physically interact with AP-1 family members. Recent major advances in artificial intelligence (AI) have resulted in great predictive power of protein-protein interactions (https://www.deepmind.com; https://alphafold.ebi.ac.uk/)[53,54]. Taking advantage of these approaches, we docked the predicted ANKRD1 3D structure to the experimentally determined crystal structure of the c-JUN/c-FOS AP-1 dimers bound to the DNA[55]. Using the HADDOCK software (https://wenmr.science.uu.nl/haddock2.4/)[56], we found several possible configurations, all of which with a high probability of ANKRD1 forming a clamp around the leucine zipper regions (bZip) of the c-JUN-c-FoOS dimer (Fig. 6c, d). Further analysis by 3DBionotes software (https://3dbionotes.cnb.csic.es/ws) pointed to several predicted interacting residues of ANKRD1 with c-JUN residues, with no predicted interactions with the corresponding part of the c-FOS protein (Fig. 6e).

The main tenets of this model were tested by direct in vitro binding assays with recombinant purified proteins. GST-tagged ANKRD1 was admixed with either c-JUN or c-FOS proteins in isolation and in combination, plus/minus the addition of an AP-1 binding DNA oligonucleotide or a cyclic decapeptide (T-5224) that interacts with the DNA binding domains (DBD) of these proteins suppressing their DNA binding activity[57]. GST-tagged ANKRD1 pulldown with a glutathione resin followed by immunoblotting showed effective binding of ANKRD1 to the c-JUN protein, irrespective of whether the DNA oligonucleotide, the T-5224 peptide or c-FOS were added, with the ANKRD1/c-JUN association being lost when c-JUN was heat denatured prior to the binding assay (Fig. 6f). Weaker binding of ANKRD1 to c-FOS alone was also detected, which was little affected by the various other additions (Fig. 6f). Results were confirmed in a second independent experiment, showing no binding of ANKRD1 to a truncated c-JUN protein lacking the DBD and bZip domains (Fig. 6g).

The findings were validated in cells by co-immunoprecipitation (co-IP) and proximity ligation assays (PLA) with antibodies against the c-JUN and FOSL2 protein, a FOS family member with a prominent role in fibroblasts[58,59] and among the most highly expressed in our own data sets of HDFs and CAFs (Supplementary Data file 7). ANKRD1 was found to associate with both proteins in HDFs overexpressing ANKRD1 as well as in CAFs (Fig. 7a–c, Supplementary Fig. 6a). The presence of ANKRD1 is of functional significance as the c-JUN-FOSL2 association, which is critically required for their activity, was drastically increased in HDFs with ANKRD1 overexpression (Fig. 7d). At the same time, it was reduced in CAFs with *ANKRD1* gene silencing (Fig. 7e). In parallel, ChIP assays showed that binding of c-JUN to the promoter/transcription regulatory regions of multiple CAF effector genes was strongly increased in HDFs with ANKRD1 overexpression (Fig. 7f, Supplementary Fig. 6b).

The binding of ANKRD1 to target genes may be mediated by its association with AP-1. As mentioned, T-5224 is a cyclic decapeptide that binds to the DNA binding domains (DBD) of c-JUN and c-FOS, suppressing their DNA binding activity[57]. ChIP assays showed that c-JUN binding to CAF effector genes was suppressed by treatment of HDFs with this compound (Supplementary Fig. 6c). Similar suppression of ANKRD1 binding to these genes was observed in ANKRD1-over-expressing HDFs and in CAFs by treatment with T-5224 (Fig. 7g, h). Reflecting the biochemical effects, treatment with T-5224 suppressed expression of CAF effector genes in the ANKRD1 overexpressing cells (Fig. 7i) as well as in CAFs (Supplementary Fig. 6d) and, in co-culture assays, was sufficient to counteract the growth-enhancing effects exerted by these cells on

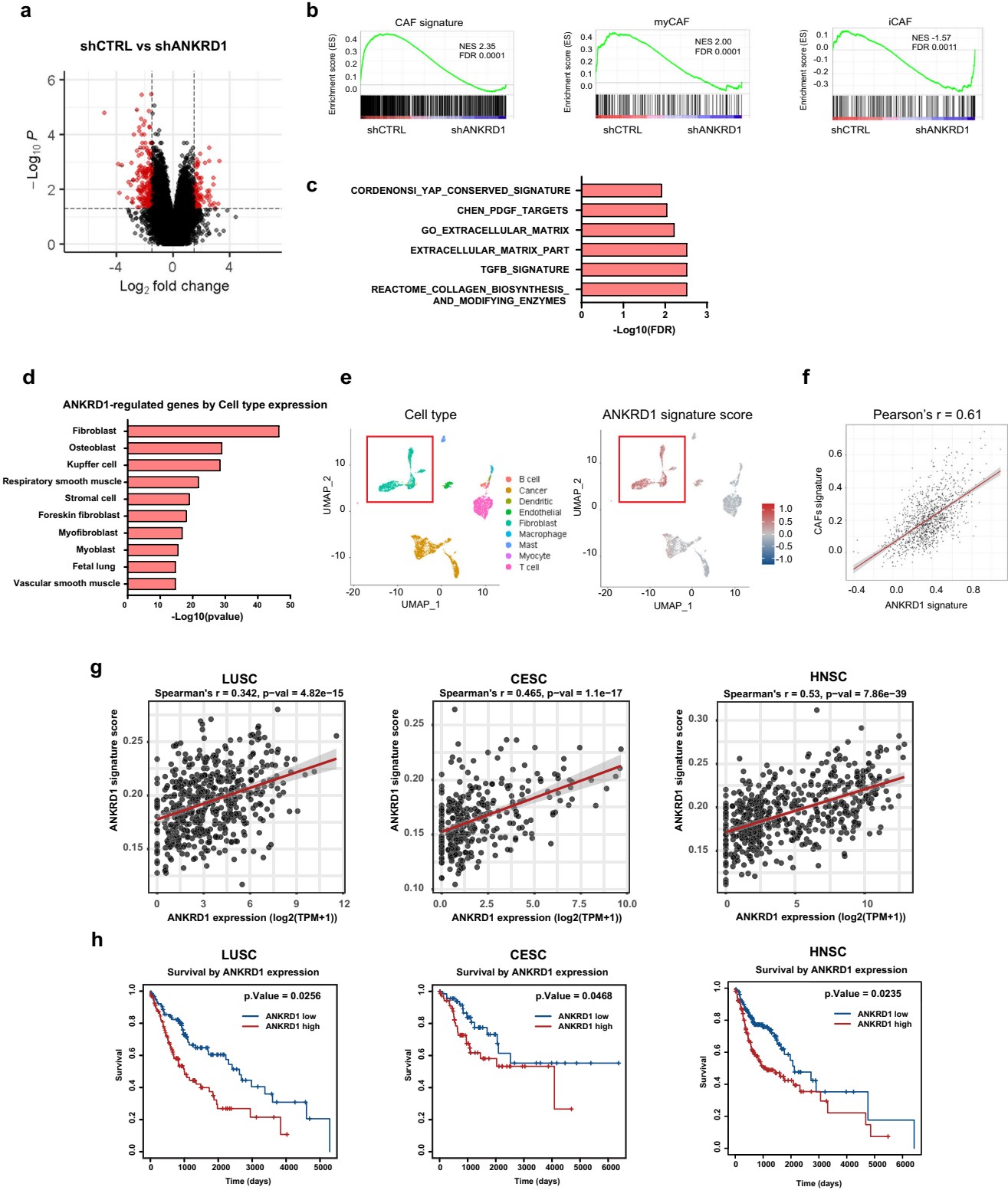

neighboring cancer cells (Supplementary Fig. 6e, f). Interestingly, treatment of either CAFs or FaDu cells with T-5224 did not affect their proliferation as assessed by Incucyte and CellTiterGlo (Supplementary Fig. 6g, h).

Thus, there is a direct and functional interaction between ANKRD1 and AP1 family members, which is of biological significance for controlling CAF activation.

## ANKRD1 targeting reproduces the effects of AP1 inhibition on CAF activation

RNA-targeting has opened new opportunities for suppressing the function of so far undruggable transcription regulatory factors[60]. We used antisense oligonucleotides (ASOs) with a chemical modification (FANA; 2-deoxy-2-fluoro-beta-D-arabinonucleic acid) reported to enhance their activity[25] to target ANKRD1. Treatment of CAFs with

**Fig. 4 | ANKRD1 regulates a CAF transcriptional program of clinical significance. a** Four different primary CAFs were infected with two shRNA targeting *ANKRD1* (shANKRD1#1, #2) or shRNA control (shCTRL) were examined by cDNA microarray hybridization and analyzed using the Transcriptomic Analysis Console software (TAC), p-values were calculated by two-way ANOVA test. Volcano-Plot of differentially expressed genes (DEG) was generated by filtering for genes downmodulated or upregulated by shANKRD1 (FC > 1.5, p value < 0.05 and FC < −1.5, p value < 0.05, respectively). Black dotted lines separate genes filtered by FC (x-axis) and p.value (y-axis). **b** Gene Set Enrichment Analysis (GSEA) of skin CAF signature, myofibroblastic CAF (myCAF) and inflammatory CAF (iCAF) were applied to *ANKRD1* silenced and control CAFs. **c** GSEA analysis of *ANKRD1*-silenced CAF profile using deposited gene sets of CAF-related pathways. Signatures were downloaded from the GSEA web page (http://www.gsea-msigdb.org/gsea/msigdb/human/genesets.jsp), and the TGFB signature was downloaded from GSE79621. **d** Cell type expression analysis using the ENRICHR tool (https://maayanlab.cloud/Enrichr/). Analysis was performed by computing genes downmodulated by shANKRD1 (FC < −2, <0.05 p value, measured with two-way ANOVA) into an extensive collection of RNA-seq data sets available through the ARCHS4 web resource (https://maayanlab.cloud/archs4/). The bar graph shows the cell type for which the genes are enriched. Values are expressed as −Log10 (p value). All the genes regulated by ANKRD1 and expressed by the cell types identified through ARCHS4 were used to build the ANKRD1 mesenchymal signature of 269 genes. **e** Uniform manifold approximation and projection (UMAP) of scRNA-seq in head and neck SCC (HNSCC). The clusters of the different cell types are as reported in ref. 40. Red boxes indicate the fibroblasts population (left), overlapping with cells expressing ANKRD1 signature (right), 5902 cells from 18 patients were analyzed. **f** Pearson's correlation analysis of ANKRD1 signature score and CAFs signature score (GSE122372) in the scRNA-seq of HNSCC shown in (**e**). **g** Spearman's correlation between *ANKRD1* expression and ANKRD1 signature in indicated patient's cohorts derived from the TCGA database (LUSC = 485 patients, CESC = 297 patients, HNSC = 504 patients). **h** Kaplan-Meier (KM) survival analysis of indicated TCGA cohorts relative to *ANKRD1* expression. Cox regression analysis. Results are adjusted for linear variables (age, sex, and stage). P values are indicated for each tumor cohort.

specific ASOs resulted in the concomitant downmodulation of *ANKRD1* and key CAF effector genes such as *ACTA2*, *COL1A1*, *INHBA*, and *HAS2* (Fig. 8a, b), while exerting no effects on their proliferation (Supplementary Fig. 7). Paralleling the effects, the association of c-JUN-FOSL2 in these cells was significantly reduced by anti-ANKRD1 ASOs treatment (Fig. 8c), and binding of the c-JUN protein to the regulatory region of CAF effector genes was strongly suppressed (Fig. 8d).

To assess whether suppression of ANKRD1 expression in CAFs by this approach exerts long term consequences on neighboring cancer cells, we pre-treated CAFs for 48 h with either ANKRD1-specific or scrambled ASOs followed by coculturing with SCC cells for seven days. SCC cancer cell expansion was significantly reduced in the presence of CAFs pre-treated with ANKRD1-ASOs relative to the controls (Fig. 8e). Even in vivo, in an orthotopic model of tumorigenesis, SCC cells admixed with CAFs pre-treated with ANKRD1-ASOs produced smaller tumors than when admixed with control CAFs, with lesser cancer cell density (Fig. 8f, g).

Thus, targeting ANKRD1 by ASOs can be a feasible approach to revert CAF activation and suppress their tumor-enhancing properties.

## Discussion

Cancer spread is the combined result of alterations of multiple cell types[61,62]. Cancer-Associated Fibroblasts (CAFs) are a major component of the tumor microenvironment that can play a primary role in cancer development[4–7]. Senescence of stromal fibroblasts is associated with early steps of CAF activation and transcriptional upregulation of a variety of genes with pro-tumorigenic functions that are also highly expressed in fully established CAFs. Therefore, targeting the transcriptional program of CAF activation without impinging on the multifaceted role of cellular senescence would be highly desirable. We previously found that the androgen receptor (AR) functions as a dual negative regulator of stromal fibroblast senescence and CAF activation[13]. We show here that ANKRD1 is a mesenchymal-specific transcriptional coactivator that connects the loss of AR function with CAF activation independently from stromal cell senescence and through increased AP-1 activity (Fig. 9). Importantly, we showed that ANKRD1 is also significantly elevated in fibroblasts from a variety of fibrotic diseases ranging from hypertrophic scarring, keloids, and idiopathic pulmonary fibrosis. The findings are of translational significance, as targeting ANKRD1 by genetic or chemical tools reverts CAF activation and inhibits cancer/stromal cell expansion.

The heterogeneity of CAF populations can result from the great plasticity of these cells, with diverging consequences on gene transcription of distinct signaling pathways. Combined functional and single-cell studies point to the importance of CAFs with myofibroblast (myCAFs) versus inflammatory (iCAFs) properties across various cancer types and mouse and human systems[2,63]. Taking a "reverse" functional approach, we showed that myCAF versus iCAF phenotypes can be induced in CAFs by activation of the TGFβ versus FGF signaling pathways[17], with further single-cell analysis revealing involvement of differential sphingolipid biosynthesis[18]. Consistent with the differential impact of TGFβ versus FGF signaling, we have found that expression *ANKRD1* in multiple HDF strains can be induced by the activation of one pathway and suppression by the other (Supplementary Fig. 8). Other cues have been reported to induce *ANKRD1* expression in other cell types, such as IL1α/β and TNFα[64–66], which may also contribute to its upregulation in CAFs. ANKRD1 is also strongly upregulated during the wound healing response upon activation of the YAP/TAZ pathway[67], which has also been implicated in CAF activation[68].

Downmodulation of AR expression and activity, as it can occur in photoaging skin, results in the concomitant induction of stromal fibroblast senescence and a large battery of genes connected with both myCAF and iCAF phenotypes[13]. Here we establish the *ANKRD1* gene as a direct negative target of AR, which is upregulated in HDFs and CAFs as a consequence of AR loss. Furthermore, upregulation of *ANKRD1* in this context is a mediator of a subset of the AR loss response, as it does not affect senescence and enhances a myCAF gene signature while suppressing that of iCAFs.

ANKRD1 transcription regulatory functions are still poorly understood[22,23]. By genome-wide profiling of ANKRD1 binding to chromatin, we found a highly significant coincidence of ANKRD1 binding peaks with binding sites of AP-1 transcription factors, which are critically dependent for their function on heterodimer formation[69]. Recent advances in artificial intelligence (AI) make it possible to model multiple protein-protein interactions with high confidence[53]. ANKRD1 was predicted by this approach to form a clamp around the experimentally determined 3D structure of AP-1 (JUN/FOS) dimers. This is of likely biochemical and functional significance, as recombinant purified ANKRD1 was found to bind to c-JUN and, to a lesser extent, c-FOS and, in both HDFs and CAFs, with elevated ANKRD1 levels promoting the association of c-JUN with FOSL2 and its binding to CAF effector target genes.

Conversely, treatment of HDFs and CAFs with T-5224, a decapeptide that inhibits c-JUN/AP-1 binding to the DNA, concomitantly suppressed binding of ANKRD1, pointing to a role of AP-1 transcription factors as anchor of ANKRD1 to target genes. Detailed structural studies will be required to further dissect these interactions. Irrespectively, targeting ANKRD1 by gene silencing and stabilized ASOs was sufficient to suppress AP-1 activity in CAFs and reverse their tumor-enhancing properties, making it a target of likely translational significance.

## Methods

### Primary human dermal foreskin fibroblasts

In this study, primary human dermal fibroblasts (HDFs) were extracted from the foreskin of young, healthy males aged 1 to 5 years. Samples

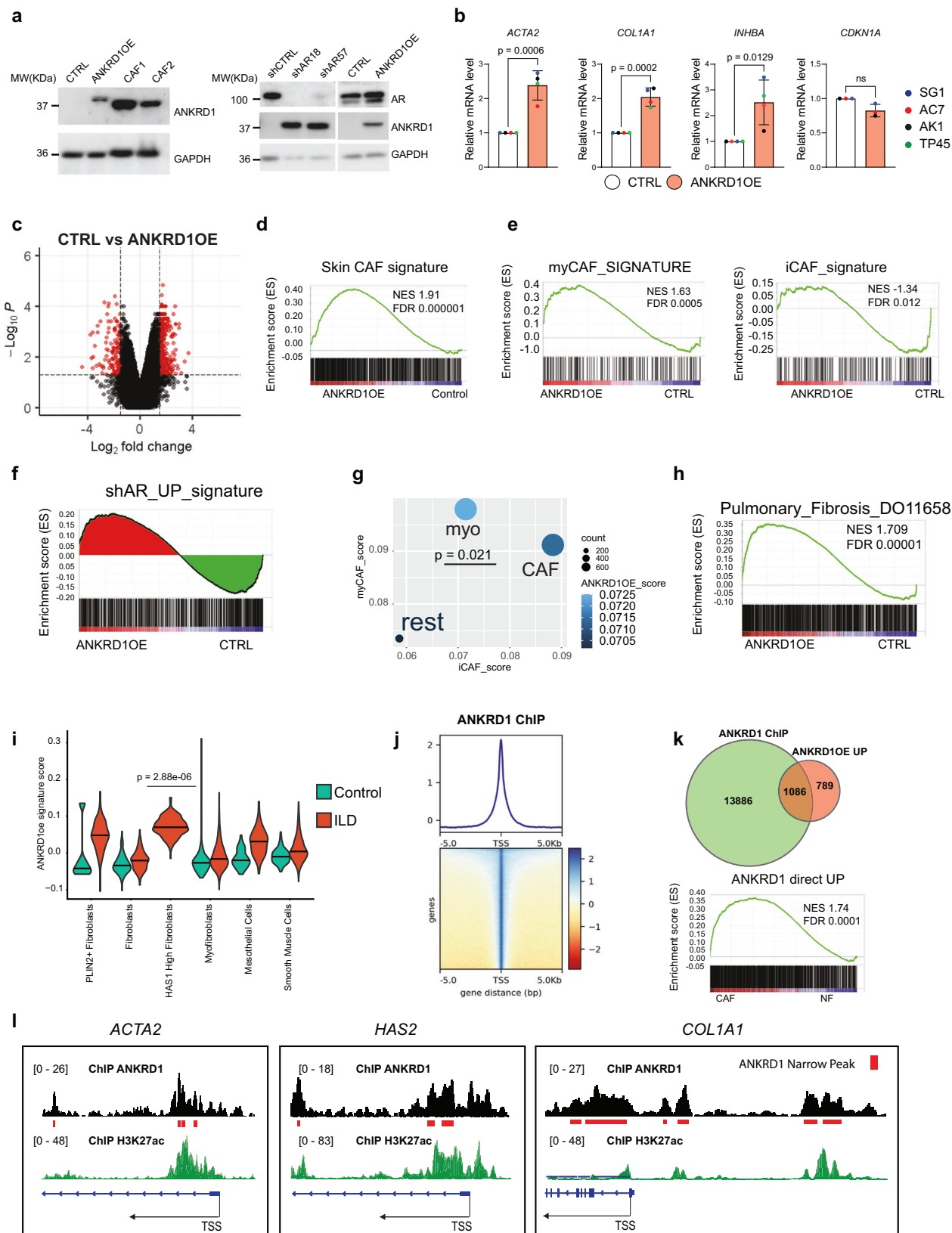

were obtained from the surgery department of the Centre hospitalier universitaire vaudois (CHUV, Switzerland), with patient and institution consent (UNIL; protocol # 222-12). Briefly, 10 mg/mL of Dispase (SIGMA) was utilized to digest the tissues. The dermis was separated from the epidermis and was then sliced into minute pieces and digested with SIGMA Collagenase (1% w/v). Next, the various

components were mixed for one hour at 37 °C. The reaction was finally stopped by adding the same volume of Dulbecco's Modified Eagle Medium (DMEM) enriched in Glutamine, Pyruvate, and high Glucose level (DMEM + Glutamax-I, Gibco) supplemented with 10% Fetal Bovine Serum (FBS, Gibco) and 1% Antibiotics (Bio Concept, Antibiotic Cocktail Penicillin-Streptomycin-Fungizone 10,000 IU/ml-10,000 ug/ml-25

**Fig. 5 | ANKRD1 is sufficient for converting HDFs to CAFs. a** Left: immunoblot analysis of ANKRD1 in ANKRD1-overexppressing (ANKRD1OE) or empty vector-control (CTRL) infected HDFs compared to two CAF strains. Right: immunoblot analysis of ANKRD1 and AR expression in HDFs infected with two shRNA targeting AR and/or shCTRL, compared to ANKRD1OE- or CTRL-infected HDFs. Anti-GAPDH was used as a loading control. Panels are derived from the same gel. The experiment was performed once. **b** RT-qPCR analysis of indicated genes in HDF strains infected with ANKRD1OE relative to CTRL infected HDFs, normalized to *RPLPO*. *n*(strains) = 4, mean ± SD, two-tailed unpaired t-test. **c** Volcano-Plot of differentially expressed genes (DEG) of three different primary HDFs infected with ANKRD1OE or CTRL vectors, generated by filtering for genes downmodulated or upregulated by ANKRD1 overexpression (FC > 1.5, *p* value < 0.05 and FC < −1.5, *p* value < 0.05 respectively, two-way ANOVA). Black dotted lines separate genes filtered by FC and *p* value. **d** Gene Set Enrichment Analysis (GSEA) was performed using the Affymetrix expression profile of HDFs infected with ANKRD1OE or CTRL vectors. **e** GSEA analysis of myCAF and iCAF signatures generated by Somerville, et al., 2020 (GSE93313). **f** Top: GSEA of a gene signature derived from AR silenced HDFs (shAR_UP_signature, GSE107321) was used in the ANKRD1OE profile. The bimodal distribution of AR signature was marked in red for the genes enriched in ANKRD1OE (shAR_UP_ANKRD1OE) and in green for the genes enriched in CTRL (shAR_-UP_CTRL). **g** Pseudo-bulk points using signature averages over fibroblast sub-populations defined by the authors cluster annotations:[40] CAF = Cancer-associated fibroblast, myo = myofibroblast, rest = resting fibroblast. The plot shows average signature scores of the iCAF signature (x-axis) versus the myCAF signature (y-axis). Size of the points represents size of the cluster in terms of cell counts. The color gradient shows average ANKRD1 scores extracted from the top 250 up-regulated genes in ANKRD1 overexpressed cells. Student's *t*-test. **h** GSEA was performed using a gene signature of pulmonary fibrosis derived from the webtool HARMONIZOME (https://maayanlab.cloud/Harmonizome/) under the MeSH ID: D011658. **i** Violin plots showing the expression of the ANKRD1 signature score in control and ILD derived mesenchymal cells from Habermann et al.[47] (GSE135893). Center line= median. Student t.test. *P* value is shown for HAS1+ population compared to all other control fibroblasts. **j** Heatmap displaying read count per million for ANKRD1-ChIP-seq peaks around the TSS (+/− 5 kb). Analysis was performed using deeptools for ChIP analysis (https://deeptools.readthedocs.io/en/develop/). **k** Direct targets of ANKRD1 obtained by overlapping the list of genes bound by ANKRD1 with the list of genes upregulated by ANKRD1OE (Two-Way ANOVA, FC > 1.5, *p* value < 0.05). The overlapping genes (ANKRD1 direct UP) were used as gene set for GSEA analysis in the transcriptomic profile of skin CAFs (GSE122372). **l** Illustration of ANKRD1 binding peaks to *ACTA2*, *HAS2*, and *COL1A1* promoters displayed using IGV software. Top layer: ANKRD1 binding peaks (black), bottom layer: H3K27ac peaks derived from ENCODE (https://genome.ucsc.edu/ENCODE/) were used to map histone modifications overlapping with ANKRD1 binding regions and downloaded from human dermal fibroblasts (green), human lung fibroblasts (blue), and human foreskin fibroblast (orange).

---

ug/ml). The cell pellet was centrifuged at 300 g, reconstituted in full DMEM with 10% FBS and 1% antibiotics, and plated at a density of $1 \times 10^5$ cells/10 cm dish culture. The attached fibroblasts were then thoroughly rinsed with phosphate buffered saline (PBS). Finally, fibroblasts were grown in full DMEM at 37 °C and 5% $CO_2$, and medium was changed every 48–72 hours. All tests were performed on cells from passages 4 to 10, and cell were routinely tested for Mycoplasma.

### Cancer-associated fibroblasts from skin
Cancer-associated fibroblasts were isolated from discarded skin samples of squamous cell carcinoma (SCC) (CAFs) in parallel with normal fibroblasts (NFs) from flanking skin. Adipose tissue was then surgically removed before the biopsy was sliced into pieces 1 or 2 mm in size. After that, 0.25 mg/ml of Liberase TL (Roche, Cat# 5401119001) was added, and the pieces were left to incubate at 37 °C for 40 minutes. Upon adding FBS to halt the reaction, tissues were filtered through a 70 μm sieve using a syringe. Following centrifugation at 300 *g*, the cells were cultured in full DMEM (10%FBS, 1% penicillin, and 1% streptomycin). Cultures were split at 80–90% confluence and medium is changed every 48 hours and multiple vials at p. 2 were frozen for further work. All work was done with early passage CAFs (p.3–6). Strains were routinely tested for Mycoplasma. Institutional Review Board (IRB# 2018P003156), Massachusetts General Hospital, Boston

### Cutaneous and oral SCC cell lines
SCC cells were extracted from the skin and the mouth and cultured as previously described[70]. James Rocco supplied oral SCC cells SCCO11, SCCO13, and SCCO22 (Massachusetts General Hospital, Boston, Massachusetts, USA). Genrich Tolstonog contributed donating both Cal27 and FaDu H/NSCC cells (CHUV, Lausanne, CH). James Rheinwald (Brigham and Women's Hospital, Boston, Massachusetts, United States) contributed with SCC13 cell line. The SCC cells used in the studies were grown and used in DMEM supplemented with 10% FBS and 1% antibiotics.

### Lentiviral production
Lentiviral particle manufacturing and infections were carried out as previously described[71]. Briefly, HEK 293 T cells were previously seeded at a confluence level of 60%. 24 hours later, cells were transfected using Polyethylenimine (PEI) and the specified vector in DMEM with FBS without antibiotics. Briefly, for the transfection of a 15 cm plate, 14 μg of DNA (containing the vector of interest and packaging vectors (my vector−CMG−VSV-G (4:2:1 ratio)) were combined with 42 μL pf PEI (1:3 ratio) and vigorously mixed and incubated for 15 minutes. In the meantime, cell media was replaced with DMEM and 10% FBS without antibiotics. After 15 minutes the mix was applied to cells, which were then incubated for 12 hours. 12 hours, the cells were washed, and the media was replaced with complete DMEM. After 48 hours, virus particles were collected and filtered using 0.45 μm filters. The viruses were kept in aliquots at −80 °C.

The cells were transduced by overnight incubationwith the mixture. The following morning, medium was replaced with full DMEM. After 48 hours, cells were incubated with the appropriate antibiotic, in order to select cells positively transduced. The shRNA vector sequences can be found in the Supplementary Data file 8.

### ANKRD1 Overexpression
Human dermal fibroblasts (HDFs) were infected with a lentiviral vector overexpressing human ANKRD1 protein, obtained from the CCSB-Broad lentiviral expression library. The ORF is inserted in the pLX304 vector, which expressed the protein fused with a V5-tag and contains Blasticidin as a resistance marker. The vectors used can be found in Supplementary Data file 8.

### shRNAs targeting of ANKRD1 and AR
CAFs were infected with lentiviral shRNA vectors designed by The RNAi Consortium (TRC). Two different shRNAs were directed against human ANKRD1 in the pLKO vector (Clone IDs: TRCN0000146636, TRCN0000148667). The empty pLKO.1 vector was used to produce control lentiviruses. All the vectors contained Puromycin as an antibiotic selection marker. HDFs were infected with two lentiviral shRNA vectors targeting androgen receptor (AR) (shAR#1, TRCN0000003718; shAR#2, TRCN0000003715) or with the pLKO vector. The vectors used can be found in Supplementary Data file 8.

### Small molecule inhibitor treatment
HDFs and CAFs were treated with the following compounds at the indicated concentrations: CAFs were treated for 48 hours with JQ1 (Cayman Chemical), dissolved in dimethyl sulfoxide (DMSO) at final concentration of 0.5 μM. HDFs and CAFs were treated with T-5224 provided by Toyama Chemical Co., Ltd. T-5224 was dissolved in DMSO solution and administered to the culture in vitro. Dose-response of T-5224 was assessed by treating HDFs with increasing doses: 1, 10, 20, 40 μM. HDFs were treated with UT-155 (MedChemExpress) at 1 μM for

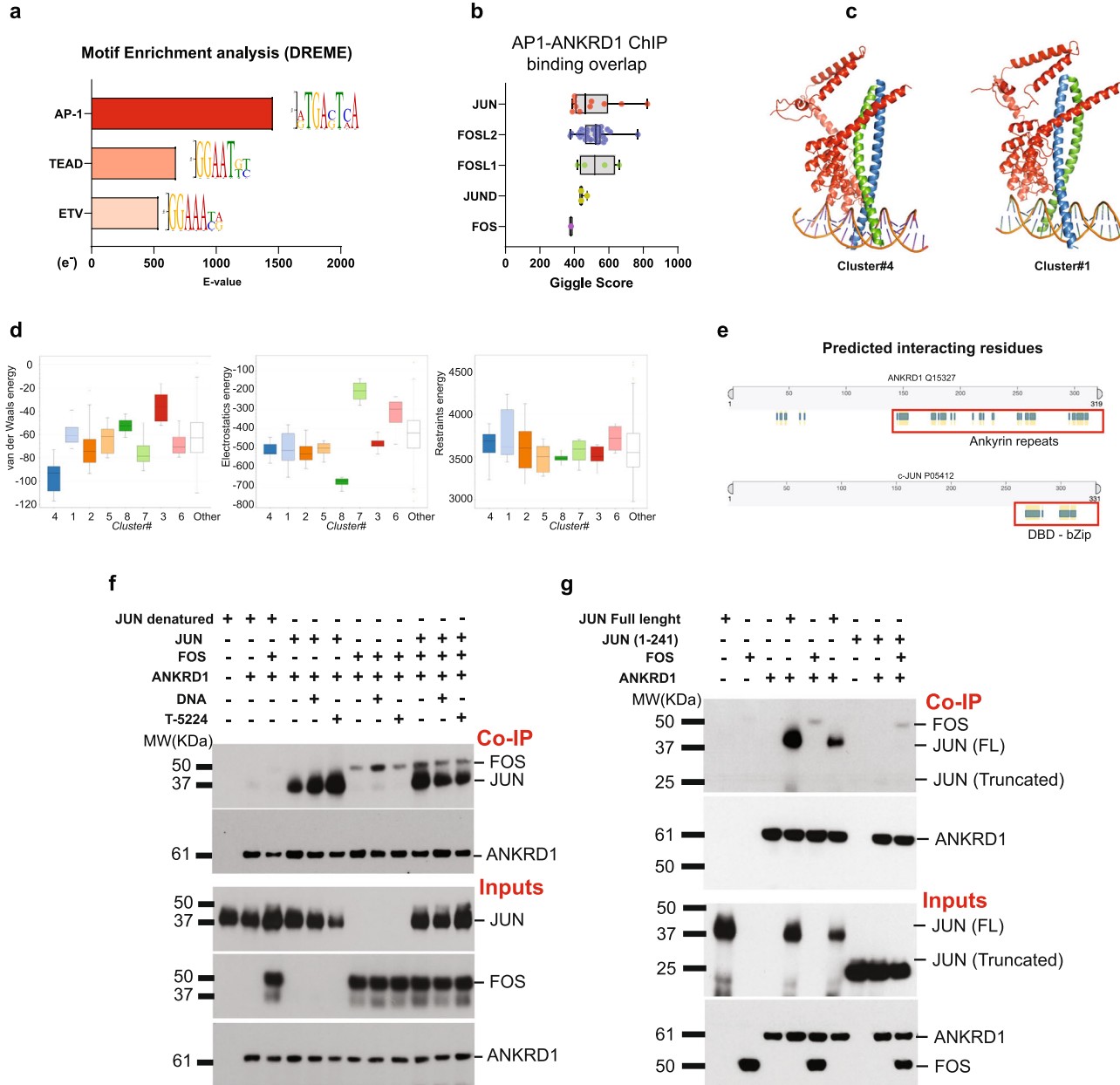

**Fig. 6 | ANKRD1 forms a complex with AP-1. a** Motif analysis of ANKRD1 ChIP-seq, assessed and quantified using MEME and DREME software (https://meme-suite.org/meme/tools/dreme). Values are expressed as log10 E-value. Shown are the top three transcription factor families enriched in ANKRD1 peak profile. **b** Prediction of transcription factors binding using the GIGGLE score. GIGGLE represents the similarity between user-defined peak profile with deposited ChIP-seq profiles in the Cistrome DB toolkit (http://dbtoolkit.cistrome.org/). GIGGLE scores for top-ranking AP1 family members on ANKRD1-bound CAF genes are represented with box plots, showing the interquartile range, and the center line representing the median value; the minimum and maximum values delineate the range of data points. Each point represents a separate ChIP-seq dataset. **c** Predicted 3D structure of ANKRD1-AP1(JUN/FOS)-DNA complex. ANKRD1 3D structure was predicted using Alphafold (https://alphafold.ebi.ac.uk/), the partial crystal structure of JUN/FOS/DNA complex was available at PDB protein databank (https://www.rcsb.org/structure/1FOS[55]). HADDOCK software (https://wenmr.science.uu.nl/haddock2.4/) was used to dock the two structures. Shown are the clusters with the lowest HADDOCK score. **d** Van der Waals energy and Electrostatics energy scores for the top eight protein clusters derived from HADDOCK docking of the ANKRD1-AP1(JUN/FOS)-DNA complex. Represented with box plots, showing the interquartile

range, and the center line representing the median value; the minimum and maximum values delineate the range of data points. **e** Schematic view of ANKRD1-JUN predicted interacting residues. The predicted ANKRD1-AP1(JUN/FOS)-DNA complex was used in 3DBionote (https://3dbionotes.cnb.csic.es/ws) for predicting the interacting residues between ANKRD1 and JUN protein. ANKRD1 is predicted to interact with JUN through the DNA-binding domain (DBD) and Leucine zipper domain (bZip) of JUN. **f** Glutathione-conjugated beads were used to immunoprecipitate GST-tagged ANKRD1 (100 ng) recombinant protein mixed with the following recombinant proteins, DNA or AP1 inhibitor: Heat-denatured JUN (100 ng), native JUN (100 ng), HIS-tagged FOS (100 ng), DNA oligo enriched with AP1 consensus motif (50 ng), or T-5224 (20 μM). Western blot analysis for ANKRD1, JUN, and FOS. The experiment was repeated once. **g** In vitro protein interactions. Glutathione-conjugated beads were used to immunoprecipitate GST-tagged ANKRD1 (100 ng) recombinant protein mixed with the following recombinant proteins: HIS-tagged JUN (truncated form 1-241aa, 100 ng), full-length JUN (100 ng), or HIS-tagged FOS (100 ng). Western blot analysis for ANKRD1, JUN, and FOS. All Co-IP proteins were run in the same nitrocellulose membrane. Similarly, all the inputs (1%) were blotted on the same membrane (also for 6f). Experiment was repeated once.

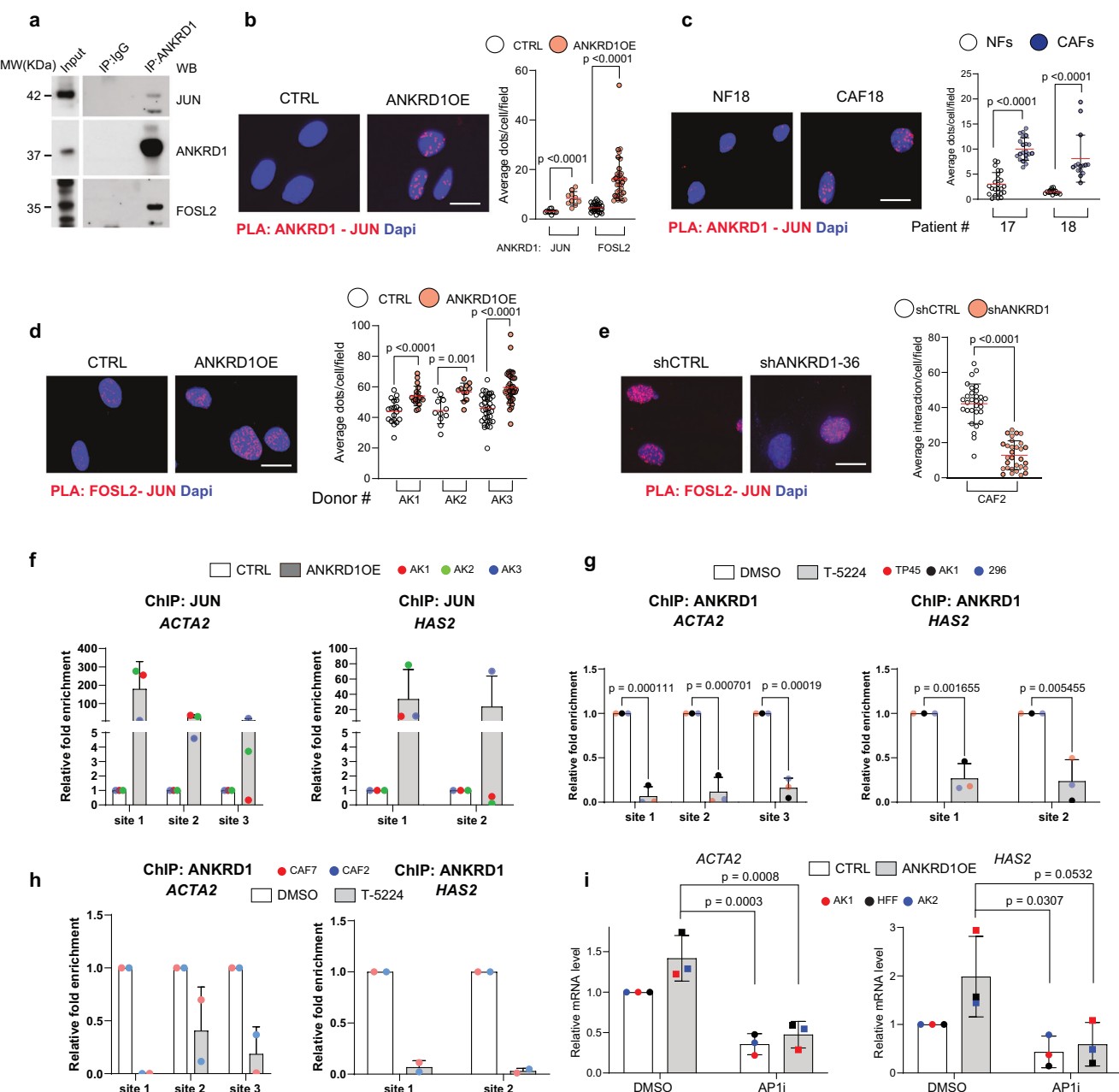

**Fig. 7 | ANKRD1 regulates CAF activation through AP-1 interaction.**
**a** Immunoprecipitation assays (IP) with anti-V5 (ANKRD1) or nonimmune (IgG) antibodies from HEK293 cells lysates infected with ANKRD1OE vector followed by immunoblotting for ANKRD1, JUN, and FRA2. IgG: nonspecific signal of IgG heavy chains. The experiment was performed twice. PLA with anti-ANKRD1 or JUN antibodies in HDFs cells infected with ANKRD1OE or CTRL vectors. For ANKRD1-JUN interaction, *n*(biological replicates) = 2, for ANKRD1-FRA2 interactions, *n*(biological replicates) = 2 (**b**) or CAFs matched with HDFs *n*(strains) = 2; (**c**). Fluorescence puncta from the juxtaposition of anti-ANKRD1 and JUN antibodies (red), DAPI (blue). Left: representative images. Right: number of puncta per cell, *n*(cells)>100 per condition, mean ± SD, unpaired two-tailed t-test. Scale bar: 20 μm. PLA with anti-JUN or FRA2 antibodies in HDFs cells infected with an ANKRD1OE or CTRL vector (**d**) or shCTRL or shANKRD1#1 vector-injected CAFs. Fluorescence puncta from the juxtaposition of anti-FRA2 and JUN antibodies (red), DAPI (blue). Left: representative images Right: number of puncta per cell, *n*(strains) = 3, *n*(cells) >100 per condition, mean ± SD, unpaired two-tailed t-test. Scale bar: 20 μm. For **d** *n*(biological replicates) = 3, **e** *n*(independent experiments) = 2 **f** ChIPmentation

analysis with anti-JUN antibody and non-immune IgGs of three ANKRD1OE or CTRL-vector-infected HDF strains (coloured dots). qPCR amplification of the indicated regions of the *ACTA2* and *HAS2* genes, expressed as relative enrichment folds over non-immune IgG in ANKRD1 overexpressing versus control HDFs. *n*(strains) = 3, mean ± SD. **g** ChIPmentation analysis with anti-V5 (ANKRD1) antibody versus non-immune IgGs of three T-5224 or DMSO-treated (48 h) ANKRD1OE–infected HDF strains (coloured dots). Results of qPCR amplification of the indicated regions for the *ACTA2* and *HAS2* genes are expressed as enrichment folds over non-immune IgGs in T-5224-treated versus DMSO controls. *n*(strains) = 3, mean ± SD, unpaired t-test with FDR multiple comparison's correction. **h** ChIPmentation analysis with anti-ANKRD1 antibody versus non-immune IgGs of two T-5224 or DMSO-treated (48 h) CAF strains. qPCR amplification of indicated regions for the *ACTA2* and *HAS2* genes, expressed as enrichment folds over non-immune IgGs in T-5224-treated versus DMSO controls. *n*(strains) = 2, mean ± SD. **i** RT-qPCR analysis of indicated genes in T-5224- or DMSO-treated (48 h) HDFs infected with ANKRD1OE or CTRL vectors, expressed relative to CTRL after housekeeping gene normalization. *n*(strains) = 3, mean ± SD. Two-way ANOVA test.

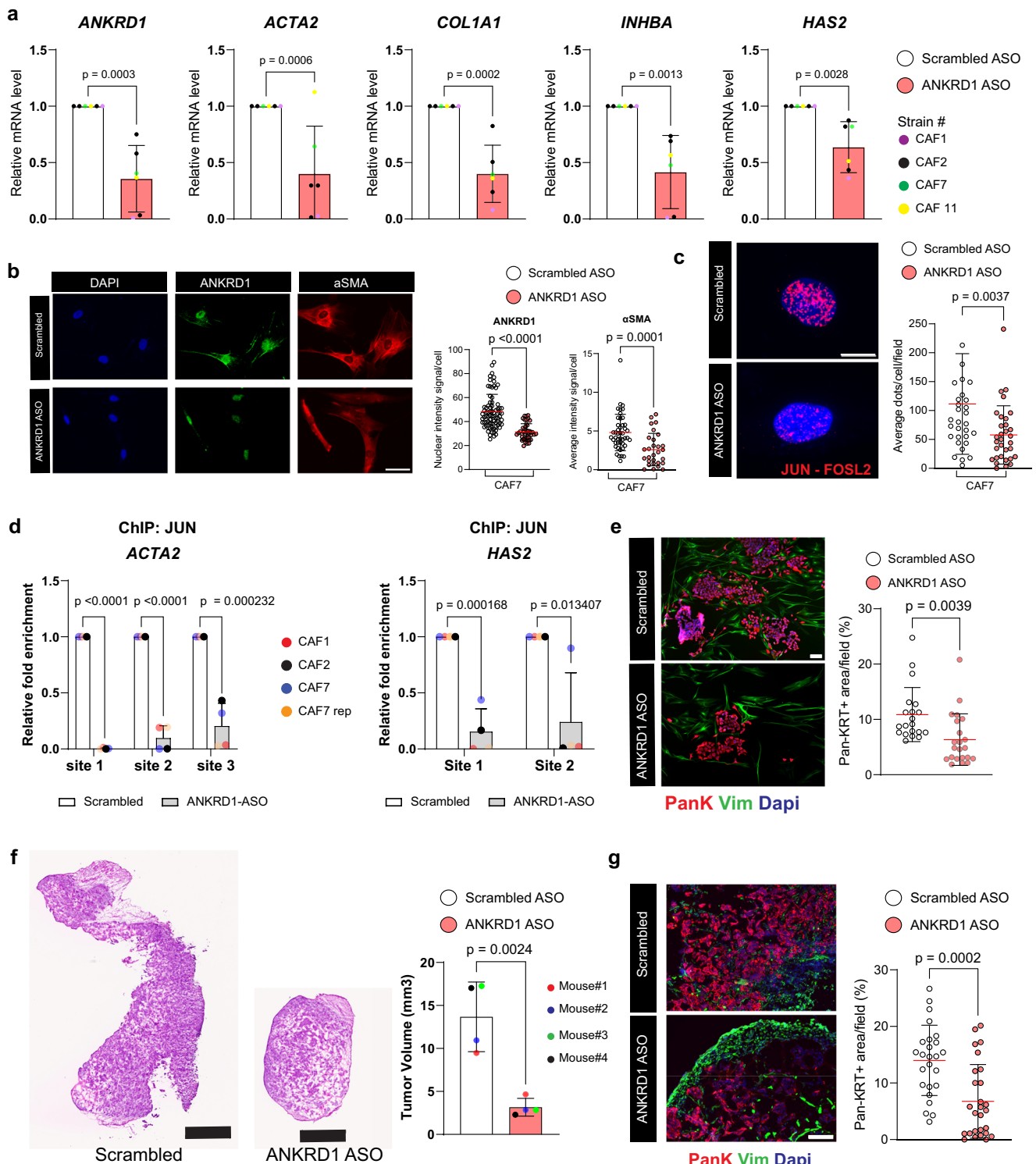

48 hours in charcoal stripped serum (FBS). Activated charcoal was provided by MilliporeSigma.

**FANA Antisense oligonucleotide (ASO) treatment**
CAFs were treated with 100 nM of ANKRD1-ASO FANA-conjugated. Briefly, cells seeded in a 6-well plate at a confluency of 30% (Corning, Fisher Scientific) were transfected with FANA-ASOs purchased from AUMBIOTECH. 10 μL of HiPerFect (Qiagen) transfection reagent, was mixed to 100 nM of FANA-ASO in DMEM only, thoroughly mixed, and incubated at room temperature for 15 minutes. After 6 hours, medium was changed with complete

DMEM. Experiments were performed after 72 hours of FANA-ASO incubation.

**Cell assays**
**EdU Incorporation.** The Click-iT EdU Alexa Fluor 647 imaging kit (Invitrogen) was employed to assess the proliferation of cells following the manufacturer's instructions. Briefly, Fibroblasts at 40% confluency were seeded on coverslips in a 12-well plate. The next day, cells were incubated for 4 hours with a 10 μM Click-iT EdU reagent. Cells were then fixed using 4% paraformaldehyde. For the co-culture assay, fibroblasts (HDFs) and cancer cells (SCC) were mixed at 1:1 ratio (2000

**Fig. 8 | ANKRD1 targeting reproduces the effects of AP1 inhibition on CAF activation. a** RT-qPCR analysis of indicated genes in multiple CAF strains transfected with 100 nM of ANKRD1-FANA or Scrambled-FANA for 72 h. Values are expressed relative to Scrambled-FANA; after house-keeping gene normalization. *n*(biological replicates) = 6, mean ± SD, two-tailed unpaired t-test. **b** Immunofluorescence images of ANKRD1 (green), αSMA (red), or DAPI (blue) in CAF#7 transfected with 100 nM of ANKRD1-FANA or Scrambled-FANA for 72 h (left) and quantification (right). *n*(independent experiment) = 3, mean ± SD, *n*(fields/condition>10), *n*(cells/field)>100, two-tailed unpaired t-test. Scale bar: 100 μm. **c** PLA with anti-JUN and FRA2 antibodies in CAF#7 transfected with 100 nM of ANKRD1-FANA or Scrambled-FANA for 72 h. Fluorescence puncta from the juxtaposition of anti-FRA2 and JUN antibodies (red), DAPI (blue), number of puncta per cell shown. *n*(independent experiment) = 2, *n*(cells) >100 per condition, mean ± SD, two-tailed unpaired t-test. Scale bar: 20 μm. **d** ChIPmentation analysis with anti-JUN antibody of three CAF strains (CAF#1, 2, 7) with an additional independent repeat of CAF7 (CAF7 rep), transfected with 100 nM of ANKRD1-ASO or Scrambled-ASO for

72 h. Results of qPCR amplification of indicated regions of the *ACTA2* and *HAS2* genes are expressed as enrichment folds over non-immune IgG in ANKRD1-ASO treated CAFs versus scrambled ASO. Results per individual CAF strains are shown as coloured dots. *n*(biological replicates) = 4, mean ± SD, multiple unpaired t-test. **e** IF for Pan-KRT (red) and VIM (green) to identify FaDu cells and CAFs, respectively. FaDu cells were co-cultured for 5 days with CAF#7 transfected ANKRD1-FANA or Scrambled-FANA (100 nM, 72 h). *n*(biological replicates) = 4, *n*(fields/condition>10), *n*(cells/field)>100. Mean ± SD, unpaired two-tailed t-test. Scale bar: 100 μm. **f** Images of H&E-stained back lesions formed by FaDu cells co-injected with CAF#2 transfected with ANKRD1-FANA or Scrambled-FANA (100 nM, 72 h) intradermally in contralateral mouse back skin, following cell embedding in Matrigel. *n*(mice) = 4, mean ± SD, unpaired two-tailed t-test. Scale bar: 500 μm. **g** FaDu cell density and proliferation, quantified as Pan-KRT positive cells per field. *n*(mice) = 4; 4–5 fields/tumor were analyzed. *n*(fields) = 25 for Scrambled and *n*(fields) = 27 for ANKRD1-ASO. Mean ± SD, unpaired two-tailed t-test. Scale bar: 100 μm.

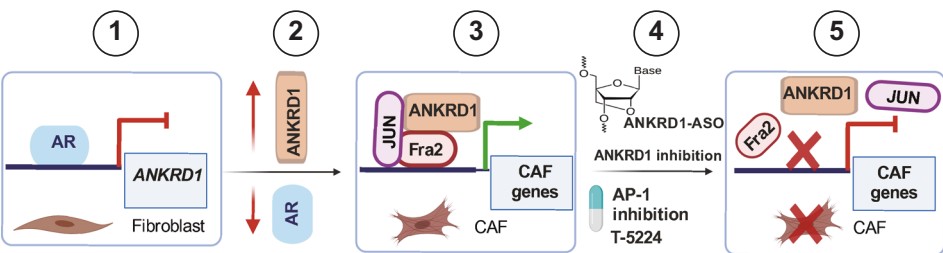

**Fig. 9 | Schematic representation of the role of ANKRD1 in CAF activation and potential translational relevance.** AR binding to the promoter of the *ANKRD1* gene and suppresses its expression (1). Downmodulation of AR at early steps of CAF activation leads to upregulation of ANKRD1 (2). ANKRD1 associates with the AP1 (JUN-FRA2) transcriptional complex and promotes AP1-dependent expression of

CAF effector genes (3). Direct targeting of ANKRD1 through stabilized anti-sense oligonucleotides (ASOs), or indirect targeting of ANKRD1 by the AP1 inhibitor T-5224 (4) disrupt the AP1 transcriptional complex and reverses CAF activation (5), suppressing cancer-stromal cell expansion. Created with Biorender.com (https://www.biorender.com/).

HDFs and 2000 SCC) on 19 mm coverslip in a 12-well plate; cells were then incubated only for 2 hours with the10 μM Click-iT EdU reagent in order to assess only SCC proliferation (higher growth ratio). Pan-Keratin staining was used to identify and quantify SCC cells that were positive for EdU incorporation. In addition, fibroblasts were stained with anti-vimentin antibody.

**Cell proliferation.** The CellTiter-Glo luminescent assay (Promega) was used to assess proliferation of cells by measuring ATP production as per the manufacturer's instructions. Dose-response curves of CAFs or FaDu cells treated with increasing doses of the T-5224 (1, 10, 20 μM) inhibitor were built using GraphPad Prism.

**IncuCyte cell proliferation.** 1000 CAFs or 1000 FaDu cells per condition were plated in quadruplicate in 96-wells plates. Cells were treated for dose-response with increasing concentration of T-5224 or DMSO. Cells were live-monitored for 7 days using the IncuCyte Zoom Live-Cell Imaging System (Essen Bioscience). For each well/ condition four images were taken every 2 hours for 1 week. The confluency of the cells over time was assessed using the IncuCyte Zoom software. The same approach was used after transfection of CAFs or FaDu cells with 100 nM of either ANKRD1-ASO or Scrambled-ASO, and cells were monitored for 7 days.

**CellTiterGlo.** Following the manufacturer's instructions, the CellTiter-Glo luminescent assay (Promega) was used to measure ATP generation for cell proliferation experiments. Experiment was carried out seeding 1000 CAFs or 1000 FaDu cells/ condition in the presence of T-5224 at different concentrations for 7 days.

**Co-culture assays**
For tumor cell expansion assays of fibroblasts-cancer cells co-culture, cells were plated onto 8-well chamber slides pre-coated with Matrigel

(BD Biosciences). In brief, chambers were coated with 100 μl Matrigel diluted 1:10 per well and incubated for 30 minutes at 37 °C to polymerize. 1000 HDF cells or CAFs were admixed with 1'000 cancer cells (FaDu or SCC13), and the combination was plated in each well. Tumor expansion was assessed five days after plating through immunofluorescence staining of Pan-keratin (Pan-K, dilution 1:300, BMA Biomedicals), Vimentin (Vim, dilution 1:300, R&D), and DAPI (dilution 1:1000) and analyzed via ImageJ.

**Spheroid formation**
Hanging-drop organoid invasion: multicellular spheroids were generated from subconfluent cells using the hanging-drop method. In brief, CAFs and/or HNSCC (FaDu) cells were resuspended in complete DMEM medium supplemented with 20% methylcellulose (Sigma Aldrich) and incubated at a final concentration of 3000 cells/25 μl drop. Drops were seeded on the internal layer of the lid of a 15 cm dish and left hanging for 72 hours. Next, pre-formed spheroids were seeded in 3D collagen lattices formed as follow: In brief, an acellular layer of collagen matrix (1.8 mg/ml collagen type I solution (Corning)/10% FBS/1x EMEM/0.03 M, L-glutamine/0.015 M NaHCO3) was first spotted on the multiwell surface plates and was allowed to pre-polymerize (37 °C, 5-10 min). Meanwhile, a mix of CAFs and HNSCC cells spheroids were washed (PBS, 2x5min) and embedded at the interphase of two layers of collagen matrix prior to collagen polymerization. Invasion type and efficacy were monitored by bright-field microscopy at day 1 and day 3 post-embedding. Image processing was performed using Fiji/ImageJ (http://fiji.sc/Fiji). The invasion was quantified as a 2D area of the invasion region from the bright field images at the above-mentioned time points. The invasion area was expressed as the difference in area between the spheroid core and the invasive area. Fusing spheroids and spheroids localized at the edge of the gel were excluded from the analysis.

Matrigel: spheroid formation was assessed by admixing 1000 HDFs or CAFs with 1000 SCC13 or FaDu cells. Cells were seeded in an 8-well chamber previously embedded with 100 μl of Matrigel. After seeding, cells were plated with additional 400 μl of DMEM with 10% serum. Sphere forming were observed for 5 days, then spheres were fixed with 4% PFA for 10 minutes and imaged with bright-field microscope.

## In Vivo experiments

Cyst assays were carried out in 6-10 weeks-old male and female NOD/SCID mice (Jackson Laboratory). $2.5 \times 10^5$ SCC13 or FaDu or SCC13 cells were admixed with an equal number of HDFs plus/minus the over-expression ANKRD1 or with ANKRD1-silenced CAFs, or with FANA-ASO treated CAFs and, after centrifugation, they were re-suspended with 70 μl of Matrigel solution (BD Bioscience) and injected intradermally in parallel into the left and right side of the mouse back. Mice were sacrificed for tissue analysis 14 days after injection, and after 10 days for the FANA-ASO experiment. Quantification of the cyst volume was assessed using the formula $V = 3.14 \times (W^2 \times L)/6$ V= volume, W= width, and L= length, the maximal tumor size permitted by the approved protocol was 20 mm in any direction, which was not exceeded in any experiment. Animal experiments were completed in accordance to the Swiss guidelines and regulations for the care and use of laboratory animals with approved protocol from the Canton de Vaud veterinary office (animal license No. 1854.4e)

## Gene and protein expression analysis

RNA isolation, cDNA synthesis, and real-time quantitative PCR (qPCR) The total amount of RNA was extracted using TRIzol per the manufacturer's instructions. cDNA was synthesized using 1 μg of mRNA and RevertAid H Minus Reverse Transcriptase (Thermo Fisher Scientific) according to the manufacturer's instructions. On a Light Cycler 480, real-time qPCR was directed using SYBR Fast qPCR Master Mix (Kapa Biosystems, Roche). The relative quantification (RQ) and expression of each mRNA were determined utilizing the comparative Ct methodology. All samples were analyzed in technical triplicate and standardized to an endogenous control, RRLP0.

## Laser Capture Microdissection (LCM) and IF-guided LCM

Microdissection was performed using an Arcturus XT microdissection system (Applied Biosystems). Eight-μm frozen tissue sections were cut and mounted on membrane-coated glass slides (LCM522, Applied Biosystems). Slides were stained in 1% methylene green (diluted in DEPC-treated water) for 10 seconds, washed three times in DEPC-treated water, and used immediately for microdissection. For RNA extraction, captured samples were collected in TRI Reagent (Sigma), and RNA extraction was performed using a standard protocol. For immunofluorescence-guided LCM, frozen blocks of normal skin and in situ SCCs were cut and fixed briefly with 75% ethanol for 30 seconds. After a brief blocking procedure (in 5% bovine serum albumin (BSA) (SIGMA) in nuclease-free PBS (GIBCO, Thermo Fisher) for 2 min), sections were incubated with a mixture of FITC-conjugated antibodies (dilution 1:1000, ThermoFisher) against PDGFRα and propidium iodide (SIGMA, P4170) for 2 min, followed by quick rinsing with PBS. The air-dried sections were then used to fluorescence-guided LCM using an Arcturus XT microdissection system as before. According to the manufacturer's recommendations, the Arcturus PicoPure RNA Isolation Kit (Applied Biosystems) was used for RNA extraction.

## Immuno Blotting

In order to estimate protein expression, cells were lysed with ice-cold RIPA buffer purchased from ThermoFisher (Catalog number: 89900), adding 50 mM NaF and protease inhibitors cocktail (ThermoFisher Scientific) for 30 minutes on ice from whole cell lysates. Protein concentration was measured using Pierce BCA protein detection kit

(ThermoFisher Scientific). Proteins from each sample were normalized to a concentration of 1 μg/1uL for proper loading conditions. After protein estimation, each sample was admixed with 2X SDS lysis buffer (Tris pH 7.5 20 mM, EDTA 1 mM, SDS 1 %) and boiled for 10 minutes at 95 °C for denaturation. Finally, samples were loaded into 8–12% SDS-PAGE gels. Separation of proteins onto the membrane was done using Trans-Blot Turbo™ Transfer System from Bio-Rad. After separation, proteins were blocked onto the membrane using 4% non-fat milk in Tris-buffered saline (TBS). Detection was established by using peroxidase-conjugated secondary antibodies (1:5000) using Super-Signal West Pico (ThermoFisher Scientific). The signals were detected on Fuji Medical X-ray films (Fujifilm).

## Co-IP from cellular lysate

Protein lysate from ANKRD1-overexpressing HEK293 cells was extracted via cell trypsinization, followed by gentle sonication and resuspension in ice-cold co-IP buffer (50 mM Tris-HCl, pH 7.4, 150 mM NaCl, 1 mM EDTA, 1% NP-40, 1% Na-deoxycholate, 0.05% SDS, and protease inhibitor (Roche)Following centrifugation, the supernatant was thoroughly digested with DNase I (ThermoFisher) to remove any lingering cell debris. 250 ug of lysate was combined with 30 uL of V5-tag magnetic beads (M167-11; MBL) in 500 uL of co-IP buffer and incubated at 4 °C for 12 hours for the Co-IP. Lysate was also incubated with rabbit IgG (as a control) for 12 hours at 4 °C. The following day, IgG-lysate was incubated with protein A magnetic beads (ThermoFisher) for one hour. The beads were washed three times with 1 ml of co-IP buffer, resuspended in SDS-PAGE sample loading buffer, and heated at 98°Cfor 20 minutes. After samples were resolved on 8% SDS-PAGE, anti-ANKRD1 (dilution 1:200, sc-365056; Santa Cruz), anti-JUN (dilution 1:1000, mAb #9165; Cell Signaling), and anti-FOSL2 (dilution 1:1000, mAb #19967; Cell Signaling) antibodies were used for immunoblotting. Abcam's VeriBlot IP Identification Reagent (dilution 1:1000, HRP; ab131366; secondary antibody) allows for the selective detection of target protein bands without background noise from denatured IgG heavy and light chains.

## Co-IP with recombinant proteins

The in vitro protein interactions using recombinant proteins were performed as follows. GST-tagged ANKRD1 (100 ng), HIS-tagged JUN (truncated form 1-241aa, 100 ng), full-length JUN (100 ng), full-length FOS (100 ng), AP-1 DNA oligo (100 ng) (sc-2501, Santa Cruz), or T-5224 (20uM) (MedChemExpress) were mixed with Glutathione-conjugated beads in binding buffer (150 mM NaCl, 100 mM Tris ph8, 0.5% NP40, 10% Glycerol) for overnight (ON) at 4 °C on a rotating platform. After this time the complexes were washed 3 times in binding buffer, then resuspended in 30 μl of SDS-PAGE loading buffer for electrophoresis and heated at 95 °C for 10 minutes. We carried out the western blots by using anti -ANKRD1, -JUN, and -FOS antibodies (reported above) To determine the requirement of a native configuration the recombinant JUN was heated for 20 min in a thermal block at 98 °C, while the same amount of ANKRD1 or FOS proteins were left on ice as control, before mixing them to the pre-heated/denatured JUN protein. Western blot was carried-out using anti-ANKRD1 (sc-365056; Santa Cruz), anti-JUN (mAb #9165; Cell Signaling), and anti-FOS (mAb #2250, Cell Signaling). The concentration/dilution of each antibody is reported in Supplementary Data File 8.

## Immunofluorescence staining and quantification

For immunofluorescence staining of cell-cultured coverslips, attached cells were first washed with cold PBS; then, cells were fixed for 10 minutes using 4% formaldehyde in PBS. Cells were then permeabilized using PBS with 0.5% Triton X-100 for 15 minutes. Cells were washed with PBS and then blocked using PBS with 2% BSA. After, cells were incubated overnight with primary antibodies previously diluted in blocking solution (dilution for each antibody

is reported in Supplementary Data File 8). The following day, cells were washed three times with PBS and then incubated with the secondary antibody conjugated with Alexa Fluor dyes (Alexa488, 568, 647, dilution 1:1000, ThermoFisher Scientific). Finally, coverslips were mounted using Dako Fluorescence Mounting Medium (Dako). The images were captured using a Zeiss LSM700 microscope. Images were quantified using Fiji software 120. For tissues, the procedure was similar: tumors were embedded in the OCT (Tissue-Tek®) compound and frozen at −80 °C, to be subsequently cryosectioned with a cryotome into 7-8 μm sections. Before staining, sections were dried for 30 minutes at room temperature and then fixed with 4% paraformaldehyde for 20 minutes. The rest of the procedure was similar as for cultured cells.

### Proximity ligation assays (PLAs)

According to the manufacturer's protocol, Proximity Ligation Assays (PLA) were performed using the Duolink PLA kit (Sigma). The following methodology was followed to conduct PLAs using a Duolink kit with the included chemicals and buffers (DUO92101; Sigma-Aldrich). Cells were plated out on glass coverslips in a 24-well plate. Cells were fixed with cold 4% paraformaldehyde (PFA) for 15 minutes at RT after being washed three times with PBS. After fixing the cells in 4% PFA, they were washed in PBS, permeabilized with 0.1% Triton X-100 in PBS for 15 minutes at room temperature, incubated in blocking buffer (supplied in the kit) for 1 hour at 37 °C in a humidified chamber, and then incubated with various primary antibodies in antibody diluents overnight at 4 °C. After being incubated with the PLA probes for 1 hour at 37 °C in a humidified environment, cells were washed three times for a total of 15 minutes in buffer A. Next, cells were incubated for 1 hour with the ligation reaction mix at 37 degrees Celsius, followed by a 3 × 5-minute wash in buffer A. Finally, the reaction was incubated with the amplification mix at 37 degrees Celsius in a dark humid chamber for 140 minutes. Samples were washed with 2x buffer B for ten minutes, then with 0.01x buffer for one minute, before being processed with mounting media. We used a Zeiss LSM880 confocal microscope to take pictures of DAPI-stained cells. Through fluorescent particle analysis in ImageJ, we were able to count the amount of PLA puncta (dots) per cell/nucleus

Antibodies used were anti-mouse ANKRD1 monoclonal antibody (Cat. 365056 Santa Cruz, 1:50 dilution), anti-rabbit JUN monoclonal antibody (Cat. #9165, Cell Signaling, 1:50 dilution), anti-rabbit FOSL2 (FRA2) monoclonal antibody (Cat. #19967, Cell Signaling, 1:50 dilution)

### ChIP-seq

Primary HDFs overexpressing ANKRD1 were previously cross-linked for protein-protein interactions with Ethylene glycol bis(succinimidyl succinate) (EGS) at a final concentration of 1.5 mM for 30 minutes. Formaldehyde was added to a final concentration of 1% for 10 min at RT. The reaction was quenched using the addition of glycine (final concentration 125 mM). Cells were washed with ice-cold PBS and collected by centrifugation (400 g). Next, cells were lysed using the iDeal ChIP-seq kit (DIAGENODE) for Transcription Factors according to the manufacturer's instructions. DNA in the cross-linked chromatin was fragmented by sonication to a 100-300 bp range using Diagenode Bioraptor. Samples were precleared using the beads included in the kit and incubated overnight at 4 °C with 5 μg of 10 μl of commercially available V5 tag antibody from GENETEX targeting the V5 tag in ANKRD1 expressing fibroblasts. Non-immune controls with non-immune IgG were included. Antibody–chromatin complexes were pulled down using protein A-beads from the kit (DIAmag protein A-coated magnetic beads, DIAGENODE). Elution was performed according to the manufacturer's instructions. Chromatin was quantified using the Qubit Fluorometric Quantification Kit (ThermoFisher Scientific).

### ChIP-Seq library preparation and data analysis

Immunoprecipitated DNA for ChIP-Seq assay from HDFs was processed as for ChIP assays. As recommended by the manufacturer, a total of 10 ng DNA was used for library preparation using NEBNext® ChIP-Seq Library Prep Reagent Set for Illumina. Sequencing was carried out at the Lausanne Genomic Technologies Facility (GTF) using the HiSeq 4000. Data analysis was carried out using the web tool https://usegalaxy.org/. Briefly, Bowtie2 [https://bowtie-bio.sourceforge.net/index.shtml] was used for fastq files alignments and MACS software [http://liulab.dfci.harvard.edu/MACS/] with default parameters was used for peak detection. The Integrative Genomics Viewer (IGV) [https://igv.org/doc/desktop/] was used for a graphic illustration of ChIP-Seq peaks and ENCODE data [https://genome.ucsc.edu/ENCODE/] for information on chromatin organization.

Raw data files from ChIP-seq assays were aligned to the GRCh38 reference genome with Bowtie2[72]. MACS2 software was used for peak detection, with a q-value cutoff of 0.05[73]. Peaks were annotated and merged with the annotatePeaks.pl and mergepeaks.pl functions, available within the HOMER software[74].

### ChIPmentation

ChIP tagmentation (ChIPmentation) and qPCR analysis were carried out using anti-JUN (5 μg, cell signaling, cat# 9165) and anti-ANKRD1 antibodies (5 μg, Santa Cruz, cat# sc-365056), and V5-conjugated magnetic beads (MBL, cat# M167-11) versus nonimmune IgGs and input material starting from $1 \times 10^6$ HDF or CAF cells. The immunoprecipitation of sonicated chromatin was carried out using a previously described protocol. Before elution, the bead-bound chromatin was tagged with Tn5 transposase (Nextera DNA Sample Prep kit, Illumina). The "tagmented" chromatin was de-cross-linked and subjected to proteinase K (Roche) digestion. Equal amounts of recovered DNA (5 ng) were subjected to amplification with tag-specific primers. Tag-specific PCR products were diluted (1:10), and 1 μl was used as a template for qPCR to determine the enrichment of the indicated sites.

All reagents including primer sequences used for qPCR, or ChIP are listed in Supplementary Data File 8.

### Transcriptomic analysis

Publicly available datasets on CAFs were retrieved on the Gene Expression Omnibus (GEO) repository: GSE22862 (non-small cell lung cancer-associated fibroblasts), GSE29270 (breast carcinoma-associated fibroblasts), GSE46824 (colorectal cancer-associated fibroblasts), GSE38517 (HNSCC), E-MTAB-2509 (Keloids), GSE40839 (IPF). ANKRD1 differential expression was evaluated with the moderated t-statistic, available within the limma package 125, for two-class comparison (CAF vs. normal fibroblasts). Moderated F-statistic was used for the three-group comparison (metastatic CAFs, primary CAFs, normal fibroblasts). *P* values were adjusted for multiple testing by using the Benjamini-Hochberg correction.

### Microarray analysis

RNA extraction and purification was done using Direct-zol RNA Miniprep Kit (Zymo Research). GeneChip® WT PLUS Reagent Kit (ThermoFisher Scientific) was used. RNA samples were checked for purity (OD260/OD280 ≥ 1.8, RIN ≥ 8), and hybridized to the human Clariom™ D Arrays (Thermo Fisher Scientific). Data were analyzed using the Transcriptome Analysis Console (TAC) software (Thermo Fisher Scientific), carried out at the Institute of Genetics and Genomics of Geneva (iGE3). Transcriptome Analysis Console (TAC) Software (Thermo Fisher Scientific) was used for data processing and analysis. Gene classification was assessed using DAVID software and ENRICHR online software[75]. In addition, Gene Set Enrichment Analysis (GSEA) was used for the analysis of global transcriptomic data using curated gene signatures obtained from the MSigDB v7.2 database (https://software.broadinstitute.org/cancer/software/gsea).

## TCGA and single-cell RNA sequencing analysis

Analyses of Head and Neck Squamous Cell Carcinoma patient datasets Normalized bulk transcriptomic profiles of 520 Head and Neck Squamous Cell Carcinoma (HNSC) patients generated with RNA-seq by The Cancer Genome Atlas (TCGA) consortium were downloaded from the GDC data portal (https://portal.gdc.cancer.gov/), along with clinical information. Data were processed and normalized by TCGA as described in the corresponding article[76]. The scores of ANKRD1 gene signature for each sample were computed using the singscore[77] R package (v1.0.0) with default parameters. Proportions of Cancer-Associated Fibroblasts (CAFs) were computed using the EPIC 130 R package (v1.1.5) with default parameters. Survival analyses were performed using the R package survival (v2.44-1.1), with *p* values computed using the log-rank test. Normalized single-cell RNA-seq profiles generated with Smart-seq2 for 18 HNSC patients by Puram and colleagues[40] were downloaded from Gene Expression Omnibus (GSE103322). Data were aligned, normalized as Transcripts Per Million, and annotated with main cell types in the corresponding article. Scaling and dimensionality reduction with PCA and UMAP were performed with Seurat v4.0.4 132. Signatures of CAFs and normal fibroblasts were extracted as genes up-and down-regulated in CAFs vs. normal fibroblasts, respectively. Signature scores for CAFs, normal fibroblasts and ANKRD1 gene signatures were computed with Seurat function 'AddModuleScore'[78]. All analyses, correlations, and statistical tests were implemented in R v4.1.1. Similar analysis was performed on single-cell RNA-seq derived from lung fibrotic fibroblasts GSE135893 to score for ANKRD1OE signature.

## Statistical significance

All statistical tests were performed with GraphPad Prism 9. (GraphPad Software, Inc.). As noted in the legends, data are shown as mean ± SEM or mean ± SD. Exact details of the statistical methods applied to each experiment are provided in the figure legends. Unless otherwise noted, two-tailed student's t-test was used to determine the statistical significance of the differences between the two groups. When comparing more than two groups, we used one-way ANOVA with Bonferroni's correction.

## Reporting summary

Further information on research design is available in the Nature Portfolio Reporting Summary linked to this article.

## Data availability

Raw and processed datasets used for this article are available under the repository accession numbers. GSE218198 (Transcriptomic analysis of human dermal fibroblasts overexpressing ANKRD1), GSE218214 (Transcriptomic analysis of CAFs with silencing of ANKRD1), and GSE218204 (ChIP-seq analysis of ANKRD1), GSE107320 (AR ChIPmentation sequencing). Source data are provided with this paper. The remaining data are available within the Article, Supplementary Information or Source Data file. Source data are provided with this paper.

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

## Acknowledgements

We thank Sandro Goruppi for the helpful discussion and An Buckinx for the critical reading of the manuscript. This study was supported by grants from the Swiss National Science Foundation (310030B_176404 "Genomic instability and evolution in cancer stromal cells") and the NIH (R01AR039190, the content does not necessarily represent the official views of the NIH). J.I. and M.K.Y. are supported by funding from the European Union's Horizon 2020 research and innovation program under the Marie Skłodowska-Curie grant agreement No 859860. E.D.C. has been supported by a "Fellowship for Abroad 2020" from Fondazione AIRC.

## Author contributions

L.M., S.G., E.D.C., J.I., A.S., and C.S. performed experiments and analyzed the results with G.P.D. L.M., D.T., M.K.Y., and P.O. performed the bioinformatic analysis of transcriptomics with G.P.D. LM and PO conducted the bioinformatics analysis of Chip-seq experiments with G.P.D. L.M. and G.P.D. designed the study and wrote the manuscript.

## Competing interests

The authors declare no competing interests.
