## [Peer Review File · Nature Communications]

REVIEWER COMMENTS

Reviewer #1 (Remarks to the Author):

In this manuscript, Mazzeo et al. demonstrates that ANKRD1 is a mesenchymal transcription cofactor that determines the conversion of normal dermal fibroblasts into carcinoma-associated fibroblasts in skin cancer. ANKRD1 is overexpressed in a large panel of CAF when compared to the normal fibroblasts counterpart. Based on their previous finding, on series of elegant bio-informatic analysis of published data and on well controlled experimentations, both in vitro and in vivo, the authors now demonstrate that ANKRD1 is both sufficient and necessary for CAF activation. This manuscript demonstrates the molecular mechanisms of ANKRD1-dependent fibroblasts activation. ANKRD1 is a transcription coactivators, which together with AP-1, controls the transcriptional regulation of CAF markers (ACTA2, COL1A1 or HAS2) as well as the pro-tumorigenic activities of CAF in vitro (proliferation and invasion of skin SCC cells) and in vivo (intra-dermal tumoral growth of skin SCC cells co-injected with CAF in which ANKRD1 expression is modulated – over-expression and knock down).

The originality of the findings provided in this manuscript makes it a strong candidate for publication in Nature Communications.

1. Several experiments within the manuscript lack error bars and statistical significance. Could the authors explain why, if there is a valuable reason, or could they please recapitulate the experiments to increase the N and thus add statistics (F1B, F1F, F2B, F2E, F3F, F7G, F8D).

2. To define the role of ANKRD1 in the protumorigenic activity of CAF, the authors set up an in vitro assay of skin SCC cells proliferation in response to CAF using a co-culture experiment. It results that depletion of ANKRD1 expression in CAF reduces the growth of FaDu cancer cells. The images showed in F3C are misleading. It seems that there is a fewer number of CAF cells in the ANKRD1 depleted samples, while it is proposed that ANKRD1 depletion does not affect KI67+ staining in CAF (F3B). I suggest the authors to re-evaluate these images and to either choose better representative images or to explain why the number of CAF is so drastically reduced in these experiments.

This comment also applies to a similar experiment shown in SF6 using the T-5224 small compound inhibitor of AP1 DNA binding domain.

3. In F8A, the authors evaluate the efficacy of chemically modified antisense oligonucleotides (ASO) to target ANKRD1. Overall, this strategy seems highly promising (F8) however, I do not understand why the authors used only two CAF strains to evaluate the expression of CAF-genes and even worse, why there are quadruplicates of CAF2 and a single point for CAF7?

Moreover, in F8F, in the tumor volume quantification, why is the mouse#4 missing for the ANKRD1 ASO?

4. A schematic representation of the findings is clearly missing at the end of the manuscript. This would bring some valuable clarity for the manuscript.

As minor comments, I found that the introduction sounds more like a discussion and it is too much based on the laboratory previous discovery.

In the methods, a clear explanation on how the normal fibroblasts isolated from patients with skin SCC have been obtained is missing.

Reviewer #2 (Remarks to the Author):

Mazzeo et al. examined the role of ANKRD1 in CAF activation and CAF-mediated tumorigenesis using in vitro and in vivo models. They found that ANKRD1 is transcriptionally upregulated in human CAFs from cutaneous squamous cell carcinoma (SCC) vs. normal fibroblasts from surrounding unaffected skin, experimentally upon silencing AR in human dermal fibroblasts (HDFs), and repressed by treatment with JQ1, which can revert the CAF phenotype through AR expression restoration. They validated this at the protein level through IF staining of human cSCC samples and demonstrate a similar upregulation in CAFs from breast cancer, lung cancer, and colon cancer, and in fibroblasts from Keloids and IPF through re-analysis of published work. Authors then show that ANKRD1 likely regulates expression of several CAF-associated genes and through co-culture assays and xenograft assays with SCC cell lines, regulates proliferative and invasive activity of SCC cell lines. Additional Co-IP, ChIPmentation and PLA assays demonstrate ANKRD1 directly binds to c-JUN and regulates c-JUN DNA-binding at CAF genes and FOSL2-JUN binding. Overall, this paper presents a lot of compelling data on the role of ANKRD1 in CAF transcriptional regulation and pro-tumorigenic functions and deepens our understanding of CAF biology. Strengths include the molecular biology data on ANKRD1 directly binding to c-JUN to coordinate expression of several CAF genes and the use of orthogonal methods to demonstrate robustness. Experiments with SCC cells would benefit from additional clarification and data. I detail some concerns below.

Major comments:

1. There is no control antibody for ChIPmentation track and quantification to assess the specificity of the AR antibody used for Fig. 2A and 2B.
2. CAFs are notoriously difficult to culture and maintain their phenotype while in culture (Puram et al. Cell 2017). Authors seemed to overcome this problem, as CAFs were used for co-culture assays and knockdown lines were generated that likely went through multiple passages. It would be important to

confirm that CAFs maintain their phenotype through all manipulations, particularly regarding maintenance of elevated ANKRD1 expression in vitro. If so, it would be further helpful if authors included methods details on which passage these cells were used or if special culture conditions were used. Current methods details only mention that cells were cultured in DMEM + 10% FBS.

3. Although Fig. 3C shows a clear SCC cell proliferation deficit upon co-culture with CAFs with silenced ANKRD1, it appears that there are fewer fibroblasts in the knockdown conditions, making it difficult to attribute the loss of proliferation directly to ANKRD1 knockdown. This also brings into question the results that suggest fibroblast proliferation is not altered by ANKRD1 KD.

4. In Fig. 3D and 3E, for CAF-tumor cell co-culture assays, there is no confirmation of CAF presence in co-culture spheroids (e.g., through IF staining). Are CAFs co-mingling with the tumor cells or forming their own spheroids? This would help establish whether the CAF-induced effects are potentially paracrine or juxtacrine in nature.

5. There is no statistical test for Fig. 3F tumor volume difference between CAF2 shCTRL and KD.

6. The same critiques mentioned above also apply to Fig. S4 (HDF ANKRD1 OE co-culture assays).

7. Do CAFs survive the in vivo injections shown in Fig. 3F and S4F? Furthermore, is KD of ANKRD1 maintained throughout the duration of the in vivo experiment?

8. Similar questions regarding length of effect for ASOs and whether there is a proliferation defect upon ANKRD1 ASO treatment.

9. All co-culture assays would benefit from comparing to the condition of SCC cells alone to understand how much the fibroblasts are aiding baseline SCC in vitro or in vivo growth, as well as invasion.

Minor comments:

1. Are full uncropped images of the Western blots available?

2. There are no methods details regarding ratio of CAFs to tumor cells used for co-culture spheroid assays.

Reply to the reviewers.

Reviewer #1:

We thank the reviewer for the positive assessment and constructive suggestions that we have addressed as follows :

1. *Several experiments within the manuscript lack error bars and statistical significance. Could the authors explain why, if there is a valuable reason, or could they please recapitulate the experiments to increase the N and thus add statistic (F1B, F1F, F2B, F2E, F3F, F7G,F8D).*

We have added calculation of the statistical significance of the results as specified below :

Fig. 1B : we have updated the figure, with an indication of the Δ CT values for each sample (after house-keeping gene normalization) and calculation of the statistical significance of the difference in ANKRD1 expression between the group of 11 CAF strains versus matched normal fibroblasts (NFs) derived from the same patients. Statistics was calculated by unpaired T test as specified in the figure legend with similar significance being obtained by paired T test. Elevated expression of CAF marker genes in CAF versus NF strains #8-16 was determined by Clariom D transcriptomic analysis of the same RNA samples as for assessment of ANKRD1 levels as shown in Suppl. Fig. 1A. Determination of CAF marker expression for CAF and matched NF strains #11 and 17, analyzed at the same passage (p3) as in our present study, was previously reported (Fig. S7 in (Capolupo et al. [1])). Elevated CAF marker expression in CAF versus matched NF strain #18 was also experimentally confirmed and can be shown as required. Stable differences in expression levels of ANKRD1 and ACTA2 were confirmed by RT-qPCR analysis of several CAF versus NF strains at 1-2 later passages (Suppl. Fig. 1B). Further determination of CAF marker expression at the protein level in CAF and NF strains # 8-16 at similar passages as in this study can be found in Fig. 2F of Katarkar et al. [2].

Fig. 1F : we have extended the IF analysis to two additional skin SCC lesions finding even in these cases significantly elevated ANKRD1 expression in CAFs versus neighbouring cancer cells (Suppl. Fig. 1E). The results were extended by statistical analysis of the LCM skin SCC samples (Fig. 1E), showing that ANKRD1 expression in 6 different uncultured CAFs differs significantly from that in fibroblasts from flanking skin, statistic was evaluated by unpaired t-test.

Fig. 2B : we have added a statistical calculation of the difference in binding of AR in 3 HDF strains to two sites of the ANKRD1 promoter with an AR recognition sequence versus a distant negative site, calculated as enrichment folds in AR immune precipitates relative to non-immune IgGs, with the control results that we now also show. Results, shown per individual HDFs strains, were combined for calculation of statistical significance by 2-way Anova test as specified in the Fig. legend.

Fig. 2E : Statistical significance was calculated by combining the results with five different HDF strains +/- AR silencing with two different shRNA vectors by 2-way Anova test as specified in the Fig. legend.

Fig. 3F : As indicated in the fig. legend, statistical significance of difference in tumor volumes has been calculated by unpaired t-test , with the exclusion of mouse n°5 (orange dot) identified as an outlier by Grubbs test ($\alpha < 0.1$). n (mice) = 4, * $p < 0.005$

Fig. 7G : we assessed ANKRD1 binding to the promoters of the *ACTA2* and *HAS2* genes in two additional HDF strains overexpressing ANKRD1 under control versus T-5224 treatment. Results, shown per individual HDF strains as coloured dots, were evaluated for statistical significance by unpaired t-test as specified in the Fig. legend.

Fig. 8D : we assessed c-Jun binding to the promoters of the *ACTA2* and *HAS2* genes +/- treatment with ANKRD1-ASO in two additional CAF strains (CAF#1 and 2) and in an independent repeat with CAF #7 . Results, shown per individual CAF strains as coloured dots, were evaluated for statistical significance by unpaired t-test test as specified in the Fig. legend.

2. To define the role of ANKRD1 in the pro-tumorigenic activity of CAF, the authors set up an in vitro assay of skin SCC cells proliferation in response to CAF using a co-culture experiment. It results that depletion of ANKRD1 expression in CAF reduces the growth of FaDu cancer cells. The images showed in F3C are misleading. It seems that there is a fewer number of CAF cells in the ANKRD1 depleted samples, while it is proposed that ANKRD1 depletion does not affect KI67+ staining in CAF (Fig. 3B). I suggest the authors to re-evaluate these images and to either chose better representative images or to explain why the number of CAF is so drastically reduced in these experiments.

This comment also apply to a similar experiment shown in SF6 using the T-5224 small compound inhibitor of AP1 DNA binding domain.

We thank the reviewer for raising this interesting point. As we now point out in the text (p. 9, line 10 from the top), we have carefully examined and quantified images of CAFs in coculture assays in multiple independent dishes per condition. While not affecting proliferation, silencing of *ANKRD1* results in consistent morphological changes of CAFs, which exhibit an elongated shape and reduced surface occupation that give the impression of reduced numbers (Suppl Fig. 3A). In fact, quantification by Image J analysis shows that ANKRD1 silencing causes no significant changes in fibroblast density, while that of SCC cells is significantly reduced (Suppl. 3B).

We performed a similar analysis for CAFs in the co-cultures of Supplementary Fig. 6E. We found that treatment with the AP1 inhibitor T-5224 caused no morphological changes. Also in this case, counting of cells by Image J analysis revealed no significant differences in the number of CAFs in cultures treated with T-5224 versus DMSO control (Supplementary Fig. 6F). To better reflect this point, the representative images in Supplementary Fig. 6E have been changed.

The work has been complemented by assessing proliferation of CAFs and cancer cells in isolation. As shown in Supplementary Fig. 6G, H, time lapse imaging and luminescence cell density

assays show that increasing concentrations of T-5224 had no effect on proliferation of either CAFs or SCC cells. The impact of T-5224 treatment on CAF marker / effector gene expression, which can explain the inhibitory effects on SCC cells in co-culture, is shown in Suppl. Fig. 6D.

3. In F8A, the authors evaluate the efficacy of chemically modified antisense oligonucleotides (ASO) to target ANKRD1. Overall, this strategy seems highly promising (F8) however, I do not understand why the authors used only two CAF strains to evaluate the expression of CAF-genes and even worse, why there are quadruplicates of CAF2 and a single point for CAF7?

In Fig. 8A, we extended the work to two additional CAF strains (CAF1 and CAF11) and show that, for a total of 4 CAF strains (CAF1, CAF7, CAF11 and CAF2 in three independent biological replicates), silencing of *ANKRD1* results in strong downmodulation of multiple CAF effector genes. Statistical significance was calculated by unpaired T test as indicated in the legend.

Moreover, in F8F, in the tumor volume quantification, why is the mouse#4 missing for the ANKRD1 ASO?

In Fig. 8F, the values for mouse 4 were already present, which were however indicated by a coloured dot overlapping with that of the experimental bar. We apologize for the confusion, which we have rectified by using different colours.

4. A schematic representation of the findings is clearly missing at the end of the manuscript. This would bring some valuable clarity for the manuscript.

We thank the reviewer for the recommendation, and have now included a summary diagram at the end of the paper (Fig. 9).

As minor comments, I found that the introduction sounds more like a discussion and it is too much based on the laboratory previous discovery.

We have re-examined the introduction to ensure that it provides general background and significance. We note that, of the 25 references in this section of the paper, only refs. 13,14 and 25 are from our laboratory.

In the methods, a clear explanation on how the normal fibroblasts isolated from patients with skin SCC have been obtained is missing.

This has now been provided.

Reviewer #2 :

We appreciate the reviewer' positive evaluation and have addressed his recommendations as follows:

1. There is no control antibody for CHIPmentation track and quantification to assess the specificity of the AR antibody used for Fig. 2A and 2B.

For Fig. 2A, we now show also the total chromatin input sequencing profile of the locus encompassing the *ANKRD1* gene (which appears as a solid line, given the low signal) and provide the CHIPmentation sequencing profile obtained after immunoprecipitation with anti-AR antibodies of a second gene, showing no specific AR binding peaks as found instead within the *ANKRD1* gene. The specificity of detection is further supported by the direct CHIPmentation analysis of Fig. 2B, showing significant enrichment with the anti-AR antibodies of two different sites of the *ANKRD1* promoter regions with AR recognition sequence relative to a negative region devoid of such sequence, or to parallel CHIPmentation results with non immune IgGs, which we also show. Results, shown per three different HDFs strains, were combined for calculation of statistical significance by 2-way Anova test values as indicated in the Fig. 2B legend.

2. CAFs are notoriously difficult to culture and maintain their phenotype while in culture (Puram et al. Cell 2017). Authors seemed to overcome this problem, as CAFs were used for co-culture assays and knockdown lines were generated that likely went through multiple passages. It would be important to confirm that CAFs maintain their phenotype through all manipulations, particularly regarding maintenance of elevated ANKRD1 expression in vitro. If so, it would be further helpful if authors included methods details on which passage these cells were used or if special culture conditions were used. Current methods details only mention that cells were cultured in DMEM + 10% FBS.

We are aware of this potential problem and for all our work, for this and other papers, we use early passage CAFs (p. 3 to 6) derived from freshly excised SCCs in parallel with HDFs from flanking skin. As specified in the method section, cultures are split at 80-90% confluence and medium is changed every 48 hours and multiple vials at p. 2 are frozen for further work. In our hands, differential expression of CAF effector genes is retained with little or no exceptions across multiple studies and different researchers handling the cells.

Elevated expression of CAF marker genes in CAF versus NF strains #8-16 was determined by Clariom D transcriptomic analysis of the same RNA samples as for assessment of *ANKRD1* levels as shown in Suppl. Fig. 1A. Determination of CAF marker expression for CAF and matched NF strains #11 and 17, analyzed at the same passage (p3) as in our present study, was previously reported (Fig. S7 in (Capolupo et al. [1])). Elevated CAF marker expression in CAF versus matched NF strain #18 was also experimentally confirmed and can be shown as required. Further determination of CAF marker expression at the protein level in CAF and NF strains # 8-16 at similar passages as in this study can be found in Fig. 2F of Katarkar et al. [2].

To further address the reviewer's concern, stable differences in expression levels of the *ANKRD1* and *ACTA2* CAF effector gene were also verified by RT-qPCR analysis of some CAF versus HDF strains at a later passage (Suppl. Fig. 1B).

3. Although Fig. 3C shows a clear SCC cell proliferation deficit upon co-culture with CAFs with silenced ANKRD1, it appears that there are fewer fibroblasts in the knockdown conditions, making it difficult to attribute the loss of proliferation directly to ANKRD1 knockdown. This also brings into question the results that suggest fibroblast proliferation is not altered by ANKRD1 KD.

As we now point out in the text (p. 9, line 6 from the top), we have carefully examined and quantified images of CAFs in coculture assays in multiple independent dishes per condition. While not affecting proliferation, silencing of *ANKRD1* results in consistent morphological changes of CAFs, which exhibit an elongated shape and reduced surface occupation that give the impression of reduced numbers (Suppl Fig. 3A). In fact, quantification by Image J analysis shows that *ANKRD1* silencing causes no significant changes in fibroblast density, while that of SCC cells is significantly reduced (Suppl. 3B).

The analysis has been complemented by assessing proliferation of CAFs upon *ANKRD1* silencing by ASO treatment. As shown in Suppl. Fig. 7, time lapse imaging assays (Incucyte) show that treatment with the anti-*ANKRD1* ASOs, like upon shRNA-mediated gene silencing, has no effect on proliferation of CAFs, while causing a parallel downmodulation of *ANKRD1* and CAF effector genes expression (Fig. 8A).

4. In Fig. 3D and 3E, for CAF-tumor cell co-culture assays, there is no confirmation of CAF presence in co-culture spheroids (e.g., through IF staining). Are CAFs co-mingling with the tumor cells or forming their own spheroids? This would help establish whether the CAF-induced effects are potentially paracrine or juxtacrine in nature.

As recommended, we have performed double IF analysis of spheroids formed by SCC cells admixed with CAFs with anti-keratin and anti-vimentin antibodies for cell type identification. We found close intermingling of the two cell types, both under control conditions and with CAFs with silenced *ANKRD1* (Fig. 3E; Suppl. Fig. 3C), as also indicated in the text (p. 9, line 13 from the bottom).

5. There is no statistical test for Fig. 3F tumor volume difference between CAF2 shCTRL and KD.

As we now indicate in the figure legend, statistical significance of differences in tumor volumes has been calculated by unpaired t-test, with the exclusion of mouse n°5 (orange dot) identified as outlier by Grubbs test ($\alpha < 0.1$). N (mice) =4, * $p < 0.005$

6. The same critiques mentioned above also apply to Fig. S4 (HDF ANKRD1 OE co-culture assays).

We have now calculated the statistical significance of the difference in volume of tumours formed by SCC cells together with ANKRD1 over-expressing HDFs versus control, as indicated in the legend for Suppl. Fig. 4A.

7. Do CAFs survive the *in vivo* injections shown in Fig. 3F and S4F? Furthermore, is KD of ANKRD1 maintained throughout the duration of the *in vivo* experiment?

We have performed double IF analysis of tumor lesions with anti-vimentin and anti-keratin antibodies for cell type identification and human-specific anti-lamin A/C for distinguishing human from mouse cells. As it is known to occur with this kind of assays [3-5], we have found that CAFs at the end of the experiment have been largely replaced with mouse cells. This is mentioned in the text and shown in the figure below, which could be added to the paper as required.

Double IF analysis of tumor lesions of Fig. 3F with anti-vimentin (VIM) and human specific Lamin A/C (hLMNA) anti- antibodies. Tumor area is delineated by dotted lines.

8. Similar questions regarding length of effect for ASOs and whether there is a proliferation defect upon ANKRD1 ASO treatment.

As shown in Fig. 8A, silencing of ANKRD1 expression (> 80% reduction) by ASO treatment of CAFs persists up to 5 days. For the *in vivo* experiments, CAFs were co-injected with SCC cells by 3 days of ASO treatment. ANKRD1 silencing in CAFs by this approach was sufficient to significantly reduce SCC cancer lesion formation (Fig. 8F). Levels of ANKRD1 expression in CAFs could not be measured at the end of the experiment since, as mentioned above, these cells were replaced by mouse cells. As requested, we have also evaluated the impact of ASO treatment on proliferation of CAFs by time lapse imaging assays (Incucyte) over 1 week, finding that proliferation of these cells was unaffected by ASO treatment (Suppl. Fig. 7), paralleling what we found with the same CAF strains upon shRNA-mediated gene silencing (Fig. 3B).

9. All co-culture assays would benefit from comparing to the condition of SCC cells alone to understand how much the fibroblasts are aiding baseline SCC *in vitro* or *in vivo* growth, as well as invasion.

As requested, for both co-culture and spheroid formation assays, we have included measurements of SCC cells (Fadu) without admixed CAFs (Fig. 3C and 3E).

Minor comments:

1. *Are full uncropped images of the Western blots available?*

These will be provided in the source file according to journal's guidelines.

2. *There are no methods details regarding ratio of CAFs to tumor cells used for co-culture spheroid assays*

This information has now been provided in the method section.

Cited literature

1. Capolupo, L., et al., *Sphingolipids control dermal fibroblast heterogeneity*. *Science*, 2022. **376**(6590): p. eabh1623.
2. Katarkar, A., et al., *NOTCH1 gene amplification promotes expansion of Cancer Associated Fibroblast populations in human skin*. *Nat Commun*, 2020. **11**(1): p. 5126.
3. Agorku, D.J., et al., *Depletion of Mouse Cells from Human Tumor Xenografts Significantly Improves Downstream Analysis of Target Cells*. *J Vis Exp*, 2016(113).
4. Cox, M.C., et al., *Tackling the tumor microenvironment - how can complex tumor models in vitro aid oncology drug development?* *Expert Opin Drug Discov*, 2023. **18**(7): p. 753-768.
5. Tentler, J.J., et al., *Patient-derived tumour xenografts as models for oncology drug development*. *Nat Rev Clin Oncol*, 2012. **9**(6): p. 338-50.

REVIEWERS' COMMENTS

Reviewer #1 (Remarks to the Author):

I would like to sincerely thank the authors for taking into account all my comments. I have no further question regarding the revised manuscript.

Reviewer #2 (Remarks to the Author):

Authors have done a commendable job on this round of revision and have addressed nearly all my concerns. My only remaining confusion lies in the interpretation of the in vivo orthotopic tumorigenesis assays (Fig. 3F-G and Fig. 8F-G). The temporal effects of human CAF co-injection on the tumor are unknown as authors have acknowledged that human CAFs are replaced by mouse cells at experiment endpoint. The human CAFs presumably die at some point during the weeks-long tumor growth and it is unknown what the effects on the cancer cells are from potentially different rates of loss of the human CAFs, which could potentially be responsible for increased tumorigenesis (instead of ANKRD1 loss specifically). An SCC cell only condition would have helped contextualize the baseline improvements in tumorigenesis induced by co-injection of human CAF. The authors have now mentioned this caveat in the text, but it's not fully explained, and IF images are referenced in the revised text and included in the rebuttal, but not actually shown in the revised manuscript, which will undoubtedly confuse readers (referencing a figure that they cannot see). I believe authors could improve discussing this caveat by acknowledging that the observed increased tumor growth in vivo during co-injection will need further follow-up work to understand how CAFs contribute to early vs. late tumorigenesis, since their experiments can only address early events (possibly cancer cell survival after implantation or early growth advantage). Authors should include the IF images of human vs. mouse-specific staining in Supplements to further clarify this point.

Minor points:

1. No references in the text to Fig. S6G-H
2. Reference to Fig. S7A should be Fig. S8 on p.18.
3. No reference to Fig. 9

Reviewer #1 (Remarks to the Author):

I would like to sincerely thank the authors for taking into account all my comments. I have no further question regarding the revised manuscript.

Reply: We appreciate the kind comment and thank the reviewer for their insightful input.

Reviewer #2 (Remarks to the Author):

Authors have done a commendable job on this round of revision and have addressed nearly all my concerns. My only remaining confusion lies in the interpretation of the in vivo orthotopic tumorigenesis assays (Fig. 3F-G and Fig. 8F-G). The temporal effects of human CAF co-injection on the tumor are unknown as authors have acknowledged that human CAFs are replaced by mouse cells at experiment endpoint. The human CAFs presumably die at some point during the weeks-long tumor growth and it is unknown what the effects on the cancer cells are from potentially different rates of loss of the human CAFs, which could potentially be responsible for increased tumorigenesis (instead of ANKRD1 loss specifically). An SCC cell only condition would have helped contextualize the baseline improvements in tumorigenesis induced by co-injection of human CAF. The authors have now mentioned this caveat in the text, but it's not fully explained, and IF images are referenced in the revised text and included in the rebuttal, but not actually shown in the revised manuscript, which will undoubtedly confuse readers (referencing a figure that they cannot see). I believe authors could improve discussing this caveat by acknowledging that the observed increased tumor growth in vivo during co-injection will need further follow-up work to understand how CAFs contribute to early vs. late tumorigenesis, since their experiments can only address early events (possibly cancer cell survival after implantation or early growth advantage). Authors should include the IF images of human vs. mouse-specific staining in Supplements to further clarify this point.

Reply: We thank the reviewer and we have added the mentioned panel in Figure S3d and discussed it in the manuscript at page 9

Minor points:

1. No references in the text to Fig. S6G-H

Reply: We apologize for the mistake and added the missing references in the text on page 16

2. Reference to Fig. S7A should be Fig. S8 on p.18.

Reply: That is correct, we apologize for the error and have edited this in the text on page 18.

3. No reference to Fig. 9

Reply: We have now referred to Fig. 9 on page 18 in the discussion.